

# Surface mass balance downscaling through elevation classes in an Earth System Model: analysis, evaluation and impacts on the simulated climate

Raymond Sellevold[1], Leonardus van Kampenhout[2], Jan T.M. Lenaerts[3], Brice Noël[2], William H. Lipscomb[4], and Miren Vizcaino[1]

[1]Geoscience and remote sensing, Delft University of Technology, Delft, the Netherlands
[2]Institute for Marine and Atmospheric Research Utrecht, Utrecht University, the Netherlands
[3]Department of Atmospheric and Oceanic Sciences, University of Colorado Boulder, Boulder CO, USA
[4]Climate and Global Dynamics Laboratory, National Center for Atmospheric Research, Boulder CO, USA

**Correspondence:** Raymond Sellevold (R.Sellevold-1@tudelft.nl)

**Abstract.** The modeling of ice sheets in Earth System Models (ESMs) is an active area of research with applications to future sea level rise projections and paleoclimate studies. A major challenge for the surface mass balance (SMB) modeling with ESMs arises from their coarse resolution. This paper evaluates the elevation classes (EC) method as an SMB downscaling alternative to the dynamical downscaling of regional climate models. To this end, we compare EC-simulated elevation dependent surface

energy and mass balance gradients from the Community Earth System Model 1.0 (CESM1.0) with those from RACMO2.3. The EC implementation in CESM1.0 combines prognostic snow albedo, a multi-layer snow model, and elevation corrections for two atmospheric forcing variables: temperature and humidity. Despite making no corrections for incoming radiation and precipitation, we find that the EC method in CESM1.0 yields similar SMB gradients as RACMO2.3, in part due to compensating biases in snowfall, surface melt and refreezing gradients. We discuss the sensitivity of the results to the lapse rate used

for the temperature correction. We also evaluate the impact of the EC method on the climate simulated by the ESM and find minor cooling over the Greenland ice sheet and Barents and Greenland Seas, which corrects a warm bias in the ESM due to topographic smoothing. Based on our diagnostic procedure to evaluate the EC method, we make several recommendations for future implementations.

## 1  Introduction

During the 20th century, the Arctic has warmed much faster than the rest of the world (e.g., Serreze and Francis, 2006; Screen and Simmonds, 2010; Hartmann et al., 2013) due to shrinking sea ice cover (Serreze and Stroeve, 2015), associated positive albedo-temperature feedbacks (Pithan and Mauritsen, 2014), and increased moisture and heat transport from the mid-latitudes (Screen et al., 2012). The Greenland ice sheet (GrIS) lies within this fragile and rapidly changing environment. The GrIS is the world's second largest ice sheet, after the Antarctic ice sheet, and has an estimated volume of $2.96 \times 10^6$ km$^3$ of ice,

leading to an increase in global mean sea level by 7.36 m if it were all melted (Bamber et al., 2013). Since the 1990s, the GrIS is losing mass at an accelerated rate (Shepherd et al., 2012; Kjeldsen et al., 2015; Mouginot et al., 2019). This mass loss is





projected to be sustained and contribute 0.04-0.21 m sea level rise by the end of the 21st century, depending on the climate scenario (Church et al., 2013). The range of uncertainty is due to uncertainties in climate scenarios, climate sensitivity and simulated mass balance of the GrIS by ice sheet models (ISMs). This latter uncertainty is currently being targeted by the Ice Sheet Model Intercomparison for CMIP6 (ISMIP6; Nowicki et al., 2016), a major international effort to investigate future ice

sheet evolution, constrain estimates of future global mean sea level and explore ice sheet sensitivity to climate forcing.

State-of-the-art Earth System Models (ESMs; coupled climate models capable of explicitly representing the carbon cycle, Flato, 2011) typically operate at a resolution of 1° (∼ 100 km), which poses a challenge for studies with a regional interest, such as GrIS SMB. For instance, the extent of GrIS ablation areas may be underestimated (Cullather et al., 2014). Downscaling techniques seem required to capture realistically the sharp gradients of SMB with elevation in the GrIS ablation zone (Lenaerts

et al., 2019). State of the art downscaling techniques are

1. Dynamical downscaling, as is done in regional climate models (RCMs, e.g., Box and Rinke, 2003; Noël et al., 2018; Fettweis et al., 2017) and recently as regional grid refinement within ESMs (van Kampenhout et al., 2018). This type of downscaling allows for explicit modeling at relatively high resolution for a region of interest. Physical parameterizations need to be readjusted over the fine grid (Hourdin et al., 2017; Schmidt et al., 2017), and in some cases, the model physics

can be better tuned for this region. A major disadvantage of this downscaling method is the computational cost and the dependency on another model for lateral forcing in the case of RCMs.

2. Statistical downscaling, which uses elevation corrections on either SMB or components of SMB (e.g., runoff). This type of downscaling is successful when realistic topographic gradients of SMB or melt are captured in the model (Helsen et al., 2012; Noël et al., 2016). However, in an ESM these gradients are typically not well-captured (Cullather et al.,

2014), making this technique unsuitable.

3. Hybrid downscaling, where elevation corrections are applied to components of SMB or surface energy balance (SEB), and the full SEB and/or SMB are explicitly calculated offline at a higher resolution. This method was used by Vizcaíno et al. (2010) to construct an SMB field from a global climate model for coupling to an ice sheet model.

A variant of the hybrid approach with "online" (that is, within the ESM) implementation has been developed recently. This

method is based on the use of elevation classes (ECs) (Fyke et al., 2011; Lipscomb et al., 2013). It simulates the SEB and SMB over glaciated surfaces, with specific albedo and snowpack evolution for each EC. This method has been successfully applied to the simulation of historical and RCP8.5-scenario projections of the GrIS SMB and mass balance evolution (Vizcaíno et al., 2013; Lipscomb et al., 2013; Vizcaino et al., 2014; Fyke et al., 2014a, b) with the Community Earth System Model version 1.0 (CESM 1.0). However, the EC downscaling in itself and its effects on the downscaled SMB and SEB components in CESM1.0

or other models have not been analyzed or evaluated before. Our study aims to fill this gap in three steps. First, we compare the simulated EC gradients of SMB and SEB components with gradients simulated by an RCM. Second, we investigate the sensitivity of the GrIS surface mass balance simulation to the main EC downscaling parameter, i.e., the temperature forcing lapse rate. Third, as the downscaling of SMB in the ECs takes place online within the climate model, we investigate how the EC implementation impacts regional climate.





Although we analyze the particular EC implementation in a specific ESM (CESM1.0), we aim to provide an evaluation and diagnostic framework to guide future implementation of EC downscaling in other climate models, for offline SMB estimates, and/or forcing of ice sheet models.

The paper is structured as follows: Section 2 describes the modeling setup as well as the regional model used for evaluation.
In Section 3 we present the results. The discussion (Section 4) addresses the strengths and limitations of the EC implementation in CESM1.0. Section 5 gives the main conclusions and outlook.

## 2  Methods

### 2.1  CESM1.0 and EC downscaling scheme

The model used for this study is the Community Earth System Model 1.0.5 (CESM1.0) (Hurrell et al., 2013) with all com-
ponents active. The atmospheric model is the Community Atmosphere Model 4 (CAM4; Neale et al., 2013) which is run at a horizontal resolution of $0.9° \times 1.25°$ and has a finite volume dynamical core. The land model is the Community Land Model 4.0 (CLM4.0; Lawrence et al., 2011) which is run at the same horizontal resolution as CAM4. Within a CLM4.0 grid cell, different land cover types can exist. The grid cell average passed to the atmosphere is calculated with an area-weighted average of the fluxes. The ocean is simulated with the Parallel Ocean Program 2 (POP2; Smith et al., 2010) with a nominal resolution of
$1°$. The ocean model grid is a dipole with its northern pole centered over Greenland to prevent numerical instabilities, implying a higher effective resolution around Greenland. Sea ice is modeled with the Los Alamos Sea Ice Model 4 (CICE4; Hunke et al., 2010; Jahn et al., 2012) which runs on the same grid as the ocean. The ice sheet model in CESM1.0 is the Glimmer Community Ice Sheet Model 1.0 (CISM1.0; Rutt et al., 2009; Lipscomb et al., 2013), with a default resolution of 5 km. For the simulations performed in this study, the GrIS ice thickness and extent does not evolve. The SMB is downscaled to a static ice sheet surface
that corresponds to present-day observations (Bamber et al., 2013).

SMB calculations in CESM1.0 are done in CLM4.0 through ECs using the CLM4.0 snowpack mass balance scheme. EC downscaling accounts for sub-grid elevation variability. SMB is explicitly calculated at multiple surface elevations to force the higher resolution ice sheet model. The EC calculation is activated in the glaciated fraction of any grid cell with total or partial glacier coverage within a pre-defined region of interest (e.g., Greenland for the present study).
The EC method takes sub-grid surface elevation from the ice sheet model and bins them into $n$ ECs. In this study, $n$ is 10 and the $n+1$ boundaries are fixed at 0, 200, 400, 700, 1000, 1300, 1600, 2000, 2500, 3000 and 10000 m elevation a.s.l. After this binning, CLM4.0 calculates the relative weight of each EC within a given grid cell, as well as the mean topography for each EC. These weights are used to calculate the grid cell average that will be used in CLM4.0, as well as for the interpolation of SMB and ice sheet surface temperature (which is equivalent to the temperature at the bottom snow/ice layer in CLM), which
are standard forcings for ice sheet models (Goelzer et al., 2013).

Through the coupling with the atmosphere model, CLM4.0 receives surface incoming shortwave and longwave radiation, precipitation, 10-m wind, relative and specific humidity, surface pressure, and 2-m air temperature. Incoming radiation, pre-cipitation, and wind are kept constant across all ECs. In contrast, the method downscales near-surface (2m) air temperature to





the ECs with a default lapse rate of 6 K km$^{-1}$, and specific humidity is downscaled by assuming the relative humidity to be constant with elevation (Lipscomb et al., 2013). At each EC, an energy balance model is used to calculate the surface energy balance (SEB; W m$^{-2}$) as

$$M = SW_{in}(1 - \alpha) + LW_{in} - \epsilon\sigma T^4 + SHF + LHF + GHF \qquad (1)$$

where $M$ is the melt energy [W m$^{-2}$], $SW_{in}$ is the incoming solar radiation [W m$^{-2}$], $\alpha$ is the surface albedo [-], $LW_{in}$ is incoming longwave radiation [W m$^{-2}$], $\epsilon$ is surface emissivity [-], $\sigma$ is the Stefan-Boltzmann constant [W m$^{-2}$ K$^{-4}$], $T$ is the surface temperature [K], $SHF$ is the sensible heat flux [W m$^{-2}$], $LHF$ is the latent heat flux [W m$^{-2}$], and $GHF$ is the subsurface heat flux into the snow or ice [W m$^{-2}$]. For these surface fluxes, positive values indicate energy transfer from the atmosphere to the land surface, and from the surface to subsurface for $GHF$. The first term on the right-hand side of Eq. (1)
is the net solar radiation, and the sum of the second and third term on the right-hand side is the net longwave radiation. As a result of the SEB calculation, CLM4.0 calculates prognostic temperature, wind, relative humidity, and other quantities, taking into account the simulated exchanges of heat and moisture and surface roughness.

Additionally, the SMB (mm water equivalent yr$^{-1}$, referred to as mm yr$^{-1}$ in this paper) is calculated at each EC as

$$SMB = SNOW + REFR - MELT - SUBL \qquad (2)$$

where $SNOW$ refers to the snowfall rate, $REFR$ is the refreezing rate of snowmelt and rainfall, $MELT$ is the sum of snow and ice melt rates, and $SUBL$ is the rate of sublimation/evaporation minus deposition/condensation. Rain and meltwater that do not refreeze are routed to runoff. For further details on the calculation of SEB and SMB, see Vizcaíno et al. (2013). Total snow mass is limited to 1 m water equivalent.

The resulting SMB is linearly interpolated onto the ice sheet grid, in two steps: first, with a bilinear horizontal interpolation
per EC, and second with a vertical linear interpolation between two ECs (above and below), based on the ice sheet model high-resolution topography.

## 2.2   Simulations design

We perform four CESM1.0 simulations with an identical setup, except for a different temperature lapse rate forcing to each of the $n$=10 ECs. These four lapse rates are 1 K km$^{-1}$, 4 K km$^{-1}$, 6 K km$^{-1}$ (default) and 9.8 K km$^{-1}$, and we refer to
the corresponding simulations as EC-1K, EC-4K, EC-6K, and EC-9.8K, respectively. EC-1K is chosen to represent minimal activation of the EC calculation. EC-4K is chosen as a lapse rate forcing between EC-1K and EC-6K that is close to the summer lapse rate over the Greenland ice sheet as estimated from observations (e.g., Fausto et al., 2009). As the upper limit of the magnitude of the lapse rate, 9.8 K km$^{-1}$ (dry adiabatic lapse rate) is used.

All simulations start in 1955 from a CMIP5 historical run that is evaluated in detail in Vizcaíno et al. (2013) (which also
describes the spinup procedure and the setup for the EC-6K) and run to 2005. The first 10 years are used for model adjustment to the new lapse rate, leaving the period 1965-2005 for analysis.



## 2.3 RACMO2.3 and evaluation procedure

For evaluation of the EC downscaled simulation of SEB and SMB, we compare with the dynamical downscaling in the Regional Atmospheric Climate Model version 2.3 (RACMO2.3; Noël et al., 2015) with a horizontal resolution of $\sim 11$ km, and forced by the ERA-Interim reanalysis (Dee et al., 2011). We analyze the period between 1965 and 2005 for both RACMO2.3 and CESM1.0. RACMO2.3 has been successfully evaluated in multiple studies by comparison with in-situ and remote sensing observations (Ettema et al., 2009, 2010; Ran et al., 2018). Version 2.3 includes updates in cloud microphysics, surface and boundary layer microphysics, radiation and precipitation (Noël et al., 2015). For the latter, precipitation falls as snow when near-surface temperatures are between -7°C and -1°C.

For the comparison, we use SEB and SMB components simulated at each EC with those simulated at the native grid of RACMO2.3. For CESM1.0, this results in between 1 and 10 values per CLM4.0 grid cell, depending on sub-grid elevation heterogeneity. We subtract each EC value of SEB or SMB component from the grid cell average, as well as the corresponding EC topographic height from the CLM4.0 mean height. With these differences, we calculate a gradient or linear function with elevation. To generate these gradients for RACMO2.3, we first cluster RACMO2.3 model output from the 11 km native grid onto the CLM4.0 grid ($\sim 100$ km). We then calculate averages for each RACMO2.3 SEB/SMB component and surface elevation over the coarse CLM4.0 grid cells. We subtract these averages from the native original values, and we construct the gradients via a linear fit. In this way, up to 56 RACMO2.3 grid cells are mapped into each CLM4.0 grid cell giving a total of 13,311 points for evaluation. For CLM4.0, the resulting number of points is 1,551.

For comparison of the overall downscaled SMB in CESM1.0 to a previous RACMO version (2.1), and an evaluation of the simulation at the mean elevation, see Vizcaíno et al. (2013).

## 3 Results

### 3.1 Process-based comparison of EC and dynamical downscaling

We use CESM1.0 output from a simulation using the default lapse rate forcing of 6 K km$^{-1}$ (EC-6K). Figure 1 illustrates the comparison of the downscaled SEB components via EC and RCM. Regression slopes $m$ (gradients) and $r$-value (correlation with elevation) are given in Table 1.

In CESM1.0, incoming solar radiation is not downscaled so that all ECs receive the same amount as simulated by the atmospheric component. In reality, however, incoming shortwave radiation generally increases with elevation as a result of thinner clouds (Van den Broeke et al., 2008; Ettema et al., 2010). RACMO2.3 simulates the incoming shortwave elevation gradient as 15.1 W m$^{-2}$ km$^{-1}$ (Table 1), giving less energy with decreasing elevation. On the other hand, for the absorbed solar radiation (Eq. 1), albedo variations generally dominate over the variations in incoming solar radiation. The albedo gradient (Fig. 1a) is underestimated in CESM1.0 (0.019 km$^{-1}$, lower albedo with decreasing elevation) when compared to RACMO2.3 (0.081 km$^{-1}$). Part of this difference may be explained through CESM1.0 not being able to capture the anomalies (-0.35 to -0.20, Fig. 1a) corresponding to very low albedos in RACMO2.3. These differences in the models arise from the treatment of




albedo during bare ice exposure. Both models treat snow albedo in a sophisticated fashion (Flanner and Zender, 2006). On the other hand, CESM1.0 and RACMO2.3 treat bare ice albedo quite differently. CESM1.0 uses a fixed value of 0.50 (0.60 for visible light and 0.40 for near-infrared radiation) while RACMO2.3 prescribes albedo from satellite observations (Noël et al., 2015), which can be as low as 0.30 for the simulated period. The albedo in RACMO2.3 is more correlated with elevation

($r$=0.60) than CESM1.0 ($r$=0.35). As a result of the underestimated gradients in both downwelling shortwave and albedo in CESM1.0, the net solar radiation gradient is also underestimated: -3.5 W m$^{-2}$ km$^{-1}$ (CESM1.0) compared to -19.6 W m$^{-2}$ km$^{-1}$ (RACMO2.3), as illustrated in Fig. 1b. In other words, the absorbed solar energy increases strongly with decreasing elevation for RACMO2.3, but only weakly for CESM1.0.

The downscaled net longwave radiation (difference between incoming and outgoing longwave radiation, Eq. 1) in CESM1.0

has an opposite gradient (8.9 W m$^{-2}$ km$^{-1}$) compared to RACMO2.3 (-3.1 W m$^{-2}$ km$^{-1}$) as shown in Fig. 1c. That is, the net longwave energy available for melting increases with lower elevation for RACMO2.3, but decreases with lower elevation for CESM1.0. The reason for this difference is that CESM1.0 does not downscale the incoming longwave radiation, while RACMO2.3 simulates a gradient of -17.6 W m$^{-2}$ km$^{-1}$ with a relatively high correlation with elevation ($r$=-0.81, Table 1). This negative correlation in RACMO2.3 is caused by thicker clouds as well as higher water vapor and atmospheric temperatures

at lower elevations (Van den Broeke et al., 2008; Ettema et al., 2010). As the outgoing thermal radiation depends on the surface temperature, both models simulate negative gradients. The result is a positive gradient for the net longwave in CESM1.0. In RACMO2.3, the magnitude of the outgoing longwave gradient is smaller than the incoming longwave gradient, resulting in a net negative gradient. Due to the complex relationship between the different components of the longwave radiation, the net longwave has a low correlation with elevation in RACMO2.3 ($r$=-0.30). In contrast, CESM1.0 simulates a high correlation

($r$=0.76) as the surface temperature gradient directly controls the net longwave gradient. The net radiation gradients in both models are 5.4 W m$^{-2}$ km$^{-1}$ (CESM1.0) and -22.6 W m$^{-2}$ km$^{-1}$ (RACMO2.3, Table 1).

In summary, biases in the downscaling of net radiation in CESM1.0 are due to null gradients of incoming radiation in the model, and weaker albedo gradients. As a result, the gradient is dominated by the outgoing longwave gradient in CESM1.0, and by the albedo and incoming longwave gradients in RACMO2.3.

Next, turbulent fluxes of latent and sensible heat are examined, as well as their contribution to the available melt energy with respect to radiation. The gradients of sensible and latent heat fluxes are negative in both models (Table 1); more energy is available for melting at lower elevation. The sensible heat flux gradient is stronger than the latent heat flux gradient and shows a larger spread of values (Fig. 1d,e.). In CESM1.0, this is a result of the elevation correction applied to the near-surface temperature (lapse rate). This correction increases atmospheric temperature and specific humidity at lower ECs and decreases

them at higher ECs within each coarse grid cell. In RACMO2.3, these heat flux gradients are smaller and less correlated with elevation ($r$=-0.42 and $r$=-0.02, for sensible and latent heat fluxes, respectively) than in CESM1.0 ($r$=-0.77 and $r$=-0.76). Stronger sensible and latent heat gradients in CESM1.0 appear to compensate for most of the underestimation of the radiation gradients (Fig. 1c,d,e.), resulting in a melt energy gradient (-16.0 W m$^{-2}$ km$^{-1}$) which is similar in magnitude and sign as RACMO2.3 (-26.1 W m$^{-2}$ km$^{-1}$; Fig. 1f, Table 1).





Figure 2 compares surface melt, refreezing, and SMB gradients between the two models. Consistent with the melt energy gradients, the surface melt gradient in RACMO2.3 is -717 mm yr$^{-1}$ km$^{-1}$ while for CESM1.0 it is -425 mm yr$^{-1}$ km$^{-1}$ (Table 1). On the other hand, the CESM1.0 refreezing gradient (62 mm yr$^{-1}$ km$^{-1}$) is in disagreement with RACMO2.3 (-129 mm yr$^{-1}$ km$^{-1}$ and Fig. 2b). CESM1.0 simulates a positive gradient, implying increasing refreezing at higher ECs despite reduced

melt rates. We hypothesize that at low ECs, this is due to limited refreezing capacity in CLM4.0, as a result of the limited snow depth (Section 2.1). On the contrary, at the higher ECs, where the melt is lower, refreezing is favored due to lower snow temperatures, more available pore space and thicker snowpacks. The overestimation of rainfall at higher elevation (Vizcaíno et al., 2013) may also be an important factor. In contrast to CESM1.0, RACMO2.3 simulates a negative gradient of -129 mm yr$^{-1}$ km$^{-1}$ (Table 1), suggesting a dominant control from the increased melting at lower elevation. As the refreezing gradient

results from the combination of opposite gradients, i.e., available meltwater and available refreezing capacity, the correlation with elevation is low in RACMO2.3 ($r$=-0.45, Table 1). It is similarly low in CESM1.0, in part due to lower correlation for the melt gradient than in RACMO2.3.

Regardless of substantial differences in melt gradients in both models, the SMB gradient is relatively close (Fig. 2c; CESM1.0: 439 mm yr$^{-1}$ km$^{-1}$ and RACMO2.3: 369 mm yr$^{-1}$ km$^{-1}$, Table 1). CESM1.0 compensates underestimation

of the melt gradient with the snowfall and refreezing gradients (in order of importance, see Table 1). While CESM1.0 does not downscale snowfall, RACMO2.3 simulates an elevation gradient of -218 mm yr$^{-1}$ km$^{-1}$ that has little correlation with elevation ($r$=0.26), possibly due to the competition of the dominant effect of height-desertification (less snowfall at higher elevations due to colder and drier air), orographic forcing of snowfall, and small scale atmospheric circulation features (Ettema et al., 2009). In addition to the snowfall contribution of +218 mm yr$^{-1}$ km$^{-1}$ to the CESM1.0 SMB gradient difference with

RACMO2.3, the difference in the refreezing gradient contributes with +191 mm yr$^{-1}$ km$^{-1}$. The higher elevation correlation of SMB with elevation in CESM1.0 ($r$=0.58) compared to RACMO2.3 ($r$=0.27) is due to the null precipitation gradient in CESM1.0.

In summary, the EC method in CESM1.0 with the default lapse rate of 6 K km$^{-1}$ (EC-6K) is approximately reproducing SMB gradients of RCM RACMO2.3. The EC method partially compensates the biases in radiation downscaling with an

overestimated turbulent heat flux gradient. The resulting melt energy gradients, however, are still lower than in RACMO2.3. However, the EC method compensates this in the net SMB gradient due to lack of snowfall downscaling (leading to a more positive gradient relative to RACMO) and a positive bias in the refreezing gradient.

## 3.2   EC downscaling sensitivity to lapse rate of temperature forcing

Figure 3 shows how the most relevant energy fluxes respond to different lapse rate forcings. With a larger lapse rate forcing,

the simulated sensible heat flux gradient is stronger, from -3.2 W m$^{-2}$ km$^{-1}$ in EC-1K to -20.0 W m$^{-2}$ km$^{-1}$ in EC-9.8K (Fig. 3 a-d). This implies that the stronger the lapse rate forcing, the more heat is redistributed from upper to lower elevations. The correlation with elevation only increases marginally when increasing the lapse rate forcing (Fig. 3 a-d).

Albedo gradients are sensitive to lapse rate forcing, from close to zero gradients in EC-1K to 0.029 km$^{-1}$ in EC-9.8K (Fig. 3 e-h). Albedo gradients are triggered by surface temperature and melt gradients resulting from turbulent heat flux gradients.





In the case of EC-1K, the turbulent heat flux gradient is not sufficient to trigger substantial albedo-melt feedback. Downscaled albedos have a variation range of similar magnitude in EC-4K and EC-6K, however more points in EC-6K have non-null variations.

The combined effects of the turbulent heat flux gradients and the associated albedo gradients result in higher melt energy gradients with higher lapse rate forcing (Fig. 3 i-l). The melt energy gradient in EC-1K is -3.5 W m$^{-2}$ km$^{-1}$ which is very similar to the sensible heat flux gradient (-3.2 W m$^{-2}$). With higher lapse rate forcings, the difference between melt energy and sensible heat gradients becomes larger, which is interpreted as an effect of the albedo-melt feedback.

The melt energy gradient as simulated by RACMO2.3 is best matched with EC-9.8K (Figure 3, Table 1). However, EC-6K matches the SMB gradient best (SMB gradients for EC-1K, EC-4K, and EC-9.8K are 110, 310 and 711 mm yr$^{-1}$ km$^{-1}$, compare with Table 1). This is explained by compensation from the snowfall and refreezing gradients.

Figure 4 compares the downscaled SMB maps on the ice sheet model grid (5 km resolution) for the four lapse rates. Spatially, the largest responses to a varying lapse rate occur along the margin of the ice sheet, and close to the equilibrium line (Fig. 4c,d). At the margins, a low lapse rate leads to a higher SMB with respect to EC-6K in a very narrow band of only 10-20 km, due to the aforementioned relatively low turbulent fluxes and weak albedo-temperature feedbacks. In the EC-9.8K, this effect becomes opposite resulting in a similarly narrow band of lowered SMB (blue rim). Further inland, this extreme lapse rate leads to larger areas with higher SMB, as higher melt energy gradients reduce melt at high ECs.

Larger lapse rates result in reduced ablation area, from 16.4% of the GrIS in EC-1K to 13.0% in EC-9.8K (Table 2. This reduction is due to an enhanced melt gradient (Fig. 3 i-l), reducing melt at higher ECs and resulting in a lower equilibrium line altitude (ELA, where SMB equals zero), and reduces interannual variability (although only mildly, from 4.0% to 3.0%). Due to this expansion of the accumulation area with higher lapse rates, the total SMB of the accumulation area increases (Table 2), although within the standard deviations. For the SMB of the ablation area, the area reduction is partially compensated with higher specific (local) ablation rates for higher lapse rates, resulting in the most negative SMB in the ablation area for EC-4K. The total SMB is the sum of the SMB for ablation and accumulation areas, and it is maximum for EC-6K. The SMB for EC-6K is at the same time the closest to RACMO2.3, also for the standard deviation. However, the range of variation of the mean total SMB across the four simulations is not large and is within the standard deviations. As an additional note of caution, the values in Table 2 result from four simulations with independent atmospheric simulation, perhaps sampling different segments of, e.g., multidecadal precipitation variability (Bromwich et al., 2001), and therefore not only reflecting the effect of the lapse rate choice.

To summarize, lapse rates lower than EC-6K result in larger ablation areas and lower integrated SMB. These results indicate a dominant effect on the CESM1.0 ELA simulation of higher melt rates at high ECs versus reduced melt rates at low ECs.

To complete this sensitivity investigation, we compare "prognostic" near-surface temperature gradients across the four simulations (Table 2). This prognostic temperature is calculated per EC within each CLM4.0 time step and is a result of heat and moisture exchange between surface and atmosphere. Therefore it differs from the prescribed lapse rate forcing. The prognostic temperature gradients are lower than the magnitude of the respective lapse rate forcing for all CESM1.0 simulations. The magnitude of the June-August (JJA) gradient is less than for December-February (DJF) and is approximately half of the forcing





lapse rate. The former is also the case for RACMO2.3. The simulation EC-9.8K gives the prognostic temperature gradient closest to RACMO2.3, which is in between the EC-6K and EC-9.8K gradients. It is remarkable that the simulation EC-4K with the lapse rate forcing that is closest to the observational summer gradient (4.7 K km$^{-1}$, Fausto et al. (2009)) and RACMO2.3 (4.3 K km$^{-1}$) is however not the simulation with the closest prognostic gradient.

## 3.3 Impact of the EC calculation on regional climate simulation

Next, we examine how the EC calculation in the land component (CLM4.0) affects the simulation of Arctic climate in CESM1.0. If the EC method is active in CLM4.0, sub-grid gradients in the ice sheet surface budget are coupled to the atmosphere model (and via the atmosphere to other components) during runtime. We compare two simulations for this analysis. The EC-1K simulation serves as the control as it represents the simulation closest to non-active EC downscaling, which is the standard for most CMIP5 ESMs. The EC-6K is used to assess the climatic effect of using the EC method. Figure 5 shows differences in selected climate variables between EC-6K and EC-1K.

Near-surface temperatures decrease over large parts of the GrIS and on average by 0.9 K in EC-6K with respect to EC-1K (Fig. 5a,b and Table 3). This relative cooling in EC-6K is due to two factors. First, because the atmospheric topography is more smoothed than the topography in the ice-sheet covered land grid cell, the atmospheric mean elevation is lower. This gives higher ECs a higher areal weight per grid cell. Second, the characteristic quasi-parabolic shape of the ice sheet contributes to this areal effect. This results in the dominance of the net (negative) energy anomalies from high ECs. Maximal cooling coincides with areas of rapid change in slope in the SE and NW. Downwind advection of colder air masses from the eastern side of the ice sheet causes mild cooling in the Greenland and the Barents Sea, which is amplified by the growth of sea-ice (Fig. 5h).

Turbulent heat fluxes respond most strongly over the Greenland ice sheet, the Labrador Sea and along the sea ice edges in Greenland and Barents Sea (Fig. 5c,d, and table 3). Significant differences over the Greenland ice sheet are collocated with areas showing a significant decrease in air temperature. In these simulations, the atmosphere transfers turbulent heat to the surface on average (Fig. 5c). The reduction in air temperature, and consequently air humidity (not shown), results in decreased turbulent heat transfer. Over the Barents Sea, larger sea-ice covered areas cause a reduction in the heat transfer from the ocean to the atmosphere.

Net surface longwave radiation increases over the Greenland ice sheet where the near-surface temperature decreases (Fig. 5f). Over these areas, incoming longwave radiation decreases; however, this is overcompensated by a reduction in emitted longwave radiation due to surface cooling.

Figure S1 compares EC-1K and EC-6K with ERA-Interim over the area in Fig. 5, with the tentative goal of assessing whether the EC method improves or deteriorates the climate simulation. However, the differences between EC-1K and EC-6K are small compared to the difference between these simulations and ERA-Interim, precluding a robust conclusion. For Greenland, on the other hand, an assessment is more reliable as the differences between the EC-1K and EC-6K simulations are of the same magnitude as with RACMO2.3. The simulation of the GrIS-averaged annual and summer near-surface air temperature is improved in EC-6K, using RACMO2.3 as a reference, as well as the net longwave radiation, melt energy, and



(only annual) turbulent heat flux (see bold values in Table 3). The simulated cooling partially counteracts the temperature overestimation in the ESM due to topographic smoothing, resulting in a close fit to RACMO2.3.

## 4   Discussion

This study has evaluated for the first time the EC method for SMB downscaling from a global climate model of $\sim$ 100 km
resolution to the much higher resolution (5 km) of an ice sheet model. This evaluation uses gradients of SEB and SMB components as a primary metric. These gradients are obtained with linear fits of all GrIS grid cells. While this provides a systematic framework of comparison, it does not account for relevant non-linear relationships for SMB gradients (e.g., Helsen et al., 2012; Noël et al., 2016) and SMB components (e.g., precipitation); or heterogeneity arising from different Greenland climate sub-regions, local influences on climate (e.g. proximity of tundra, valleys, fjords), or proximity to the ELA.

We justify our comparison with the RCM as dynamical downscaling is the most advanced downscaling technique as shown in numerous evaluations (e.g., Ettema et al., 2010; Noël et al., 2015). However, one of the limitations of comparing with an RCM is it being laterally-forced by reanalysis and has fundamental differences in the physical schemes and simulated climate components relative to the ESM. Additionally, RACMO2.3 has some well-documented biases, e.g., an underestimation of net longwave, which is compensated by the sensible heat flux (Ettema et al., 2010; Noël et al., 2015). Further, the RACMO2.3
model was forced at its lateral boundaries by ERA-Interim reanalysis (Dee et al., 2011), which limits the "intrinsic" or "natural" climate variability compared to an ESM. Therefore, a more systematic comparison could be made by forcing an RCM with the same ESM where the EC method is implemented.

As a result of the combination of EC downscaling and advanced snow physics (Lipscomb et al., 2013), CESM1.0 shows high skill in simulating GrIS climate compared to same-generation global climate models/earth system models (Cullather
et al., 2014). The ability to realistically represent GrIS SMB in ESMs has been utilized for projections of future SMB change (Vizcaino et al., 2014; Fyke et al., 2014a, b), without an RCM for additional dynamical downscaling. Reliable simulation of the GrIS surface climate at ESM resolution enables to explore the interaction with other climate components (e.g., atmosphere, ocean, sea-ice).

While the EC method in CESM1.0 realistically simulates SMB gradients, we have shown here major deficiencies in the
simulation of individual gradients of surface energy and mass balance components compared to RCM. This is an important caveat for modelers who may need to calculate the SMB from individual components of the energy or mass balance, e.g., to perform corrections for one atmospheric forcing field. It also limits the possibility to investigate individual processes at a higher resolution. In the following, we discuss the relative importance and possible fixes of the biases in these individual processes as identified for CESM1.0.

1. CESM1.0 does not capture low enough albedo values due to the use of a single fixed ice albedo, while bare ice has a broader range of albedos (Alexander et al., 2014). We recommend therefore the use of spatially varying ice albedos, e.g., to simulate the impacts of impurities on ice "darkening" (Wientjes et al., 2011; Ryan et al., 2018).



2. The EC scheme in CESM1.0 does not downscale incoming radiation, although it varies over small scales at the GrIS surface (Van den Broeke et al., 2008; Van Tricht et al., 2016). The lack of downward longwave downscaling leads to an underestimation of net radiative energy at low ECs and an overestimation at high ECs. We recommend downscaling of incoming radiation to reduce over-compensation from the turbulent heat fluxes gradients and more realistically capture
radiation-snow-ice interactions such as shortwave-generated subsurface snowmelt.

3. Since snowfall has no elevation corrections in CESM1.0, small-scale orographically induced precipitation, height-desertification effects, and small scale variations in the rain to precipitation ratio are not captured. Designing realistic and effective elevation corrections for precipitation is a challenging task as the precipitation's correlation with elevation is spatially highly variable over the GrIS (Noël et al., 2016). To account for fine-scale variations in the rain to precipitation
ratio with a simple parameterization, we propose the implementation of a scheme relating the phase of precipitation with atmospheric near-surface temperature, similarly as in Noël et al. (2015).

4. CESM1.0 does not realistically simulate the refreezing gradient, mainly due to limited snow mass in the CLM4.0 snow-pack and biased high rainfall rates at high elevations. A realistic simulation of refreezing is key in modeling the response time of an ice sheet to a changing climate (van Angelen et al., 2014) as it acts as a buffer for meltwater to run off the
ice sheet surface. A more physically based treatment of snow could be used with a snow densification scheme that does not impose a maximum allowed snow depth. An intermediate approach is using relatively large snow and firn depths. As an example along this line, the maximum snow depth can be increased, as in the version 5.0 of CLM, with respect to CLM4.0 due to the further development of the snow scheme to allow for realistic firn simulation (van Kampenhout et al., 2017).

Assessing the optimal choice of lapse rate forcing proves challenging, as certain lapse rates score better for some metrics than others. In this study, the EC-1K results in the turbulent heat flux gradients closest to RACMO2.3 (Fig. 3a), but almost null melt energy and SMB gradients. EC-4K does not stand out in any way. EC-6K results in the most realistic SMB gradients, despite EC-9.8K comparing the best with RACMO2.3 for the melt gradient. This discrepancy is because CESM1.0 does not downscale snowfall which has an opposite slope to the melt gradient. For the downscaled SMB, EC-6K and EC-9.8K give
fairly similar results, making it hard to distinguish one or the other as the best choice.

Global climate models often have warm biases over high areas like the ice sheets, due to topographic smoothing. Here we showed that the EC implementation in CESM1.0 results in moderate cooling over Greenland, which fully corrects the warm bias with respect to the RCM. The cooling pattern from the EC method is similar to that of Franco et al. (2012) who explored the sensitivity of the simulated GrIS surface climate to horizontal resolution with an RCM.

**5 Conclusions**

The EC downscaling as implemented in CESM1.0 results in realistic GrIS SMB gradients as shown through comparison with a state-of-the-art RCM. In CESM1.0, high turbulent heat flux gradients compensate for radiation, which is not downscaled.

Explicit simulation of snow albedo enables the albedo-melt feedback which is shown to contribute to realistic melt gradients and consequently realistic SMB gradients. Therefore, we conclude that the EC classes method in CESM1.0 is efficient to generate a realistic downscaled SMB, despite the fact that only temperature and humidity are downscaled.

Our sensitivity experiments reveal that a larger lapse rate for the temperature correction results in higher melt energy gradi-
ents. As a consequence of these gradients, ablation areas narrow in CESM1.0, although this result may be different for other models or ice sheet topographies. In turn, this leads to a general cooling downwind of Greenland and an increase in sea ice cover over the Greenland Sea and the Barents Sea. For future implementations of the EC classes within ESMs, we recommend evaluation of the effects on regional climate simulation.

Future improvements of the EC method could be headed towards realistic downscaling of the individual surface energy and
mass budget components. Some concrete examples include, (1) a lower and/or spatially varying albedo; (2) downscaling of incoming radiation; (3) downscaling of precipitation phase; and (4) development of more adequate snowpack parametrizations, fit for polar conditions.

This study aims to guide future implementation of the EC method, providing diagnostic metrics and evaluation methodology. We recommend in any case that these metrics are adapted to the particular targets of scientific research to be conducted with
each model.

*Code and data availability.*  The model CESM1.0.5 can be downloaded from http://www.cesm.ucar.edu/models/cesm1.0/. Code used for processing of data and plotting of results can be found at [github repository; link will be added by final manuscript]. Data from the CESM1.0.5 simulations, only the data presented here and fully processed can be downloaded from [zenodo repository; link will be added by final manuscript].

*Author contributions.*  The idea of the study and simulations design came from RS and MV. RS carried out the model simulations, data analysis and writing of the manuscript, under the supervision of MV. LK contributed to the development of the analysis software and BN provided RACMO2.3 data. All authors read and commented on the manuscript.

*Competing interests.*  The authors declare no competing interests.

*Acknowledgements.*  RS acknowledges support from the Netherlands Organization for Scientific Research (NWO) via project ALWOP.2015.096,
and MV from the European Research Council ERC-StG-678145-CoupledIceClim. LK acknowledges support from the Netherlands Earth System Science Centre (NESSC), financially supported by the Ministry of Education, Culture and Science (OCW, Grantnr. 024.002.001). BN acknowledge funding from the Polar Program of NWO and NESSC. Computing and data storage resources, including the Cheyenne supercomputer (doi:10.5065/D6RX99HX), were provided by the Computational and Information Systems Laboratory (CISL) at the National



Center for Atmospheric Research (NCAR). The material is based upon work supported by NCAR, which is a major facility sponsored by the National Science Foundation under Cooperative Agreement No. 1852977. The CESM project is supported primarily by the National Science Foundation.



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



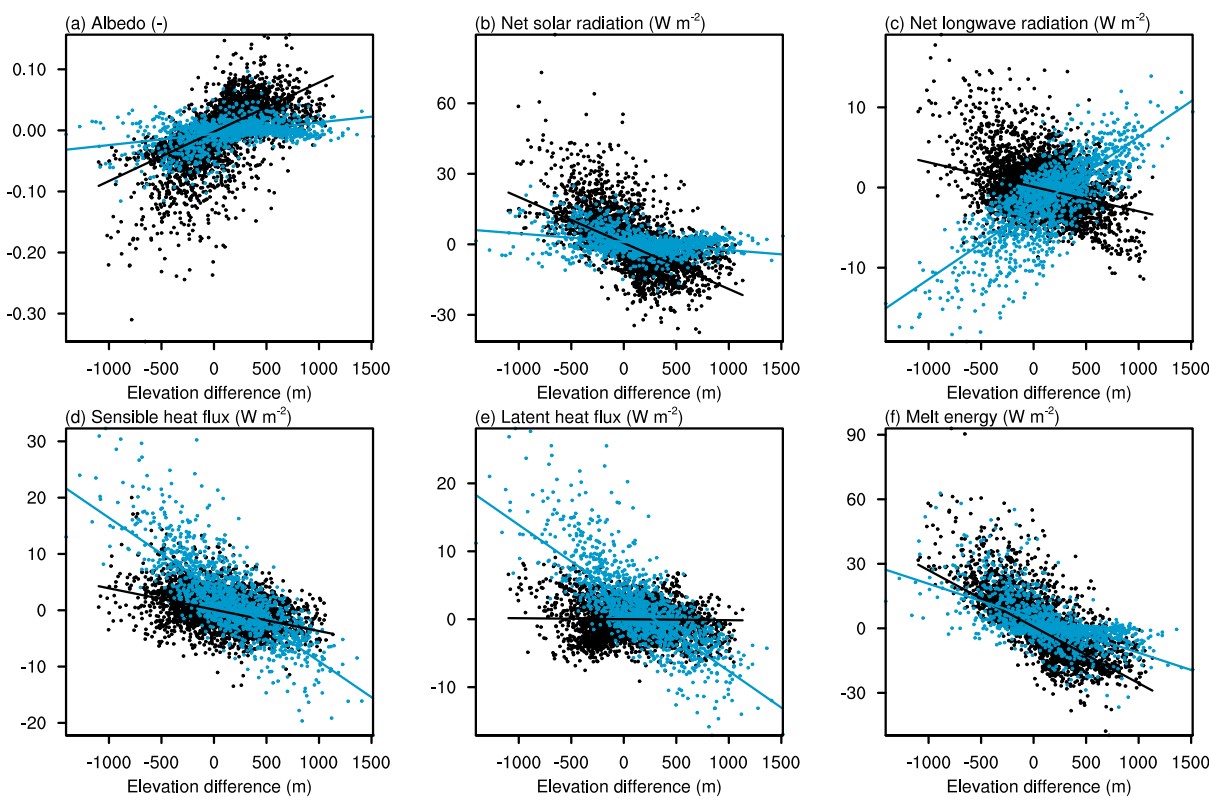

**Figure 1.** Comparison of EC downscaling (simulation EC-6K, blue) versus dynamical downscaling in a RCM (RACMO2.3, black) for several summer (JJA) SEB components and near-surface climate, a) albedo, b) net solar radiation (W m$^{-2}$), c) net longwave radiation (W m$^{-2}$), d) sensible heat flux (W m$^{-2}$), e) latent heat flux (W m$^{-2}$) and f) melt energy (W m$^{-2}$). The x values show deviation of surface elevation (m) from the coarse grid cell ($\sim$ 100 km) mean, and the y values show deviation of the physical quantity from the grid cell mean. In plots (b) through (f), positive y values indicate more energy available for melting. Melt energy (f) is the sum of the radiation and turbulent flux in terms in (b) through (e), plus the ground heat flux (not shown). The lines represent least-squares linear regression.



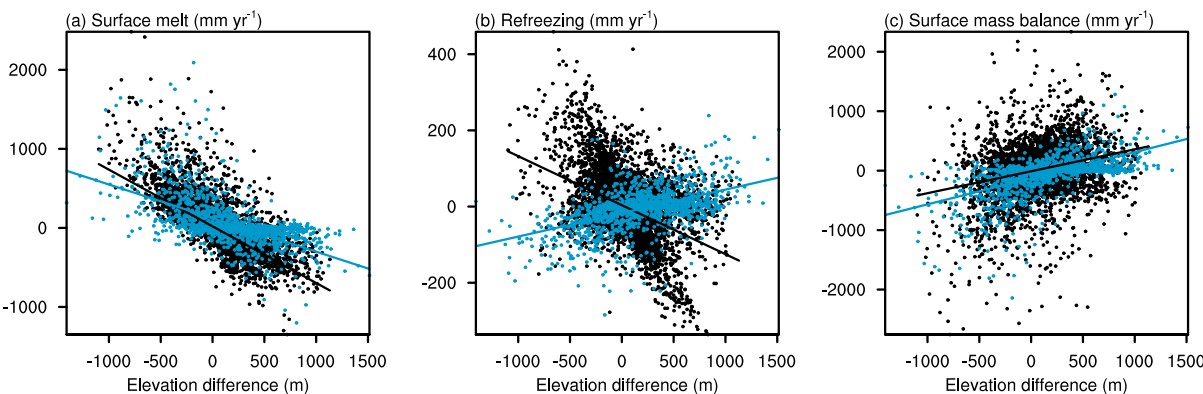

**Figure 2.** Same as figure 1, for annual SMB components from EC-6K (blue) and RACMO2.3 (black). a) Surface melt (mm yr$^{-1}$), b) refreezing (mm yr$^{-1}$) and c) surface mass balance (mm yr$^{-1}$). Surface mass balance is the sum of snowfall (not shown) and refreezing (b), minus the surface melt (a) and sublimation (not shown). The lines represent least-squares linear regressions.



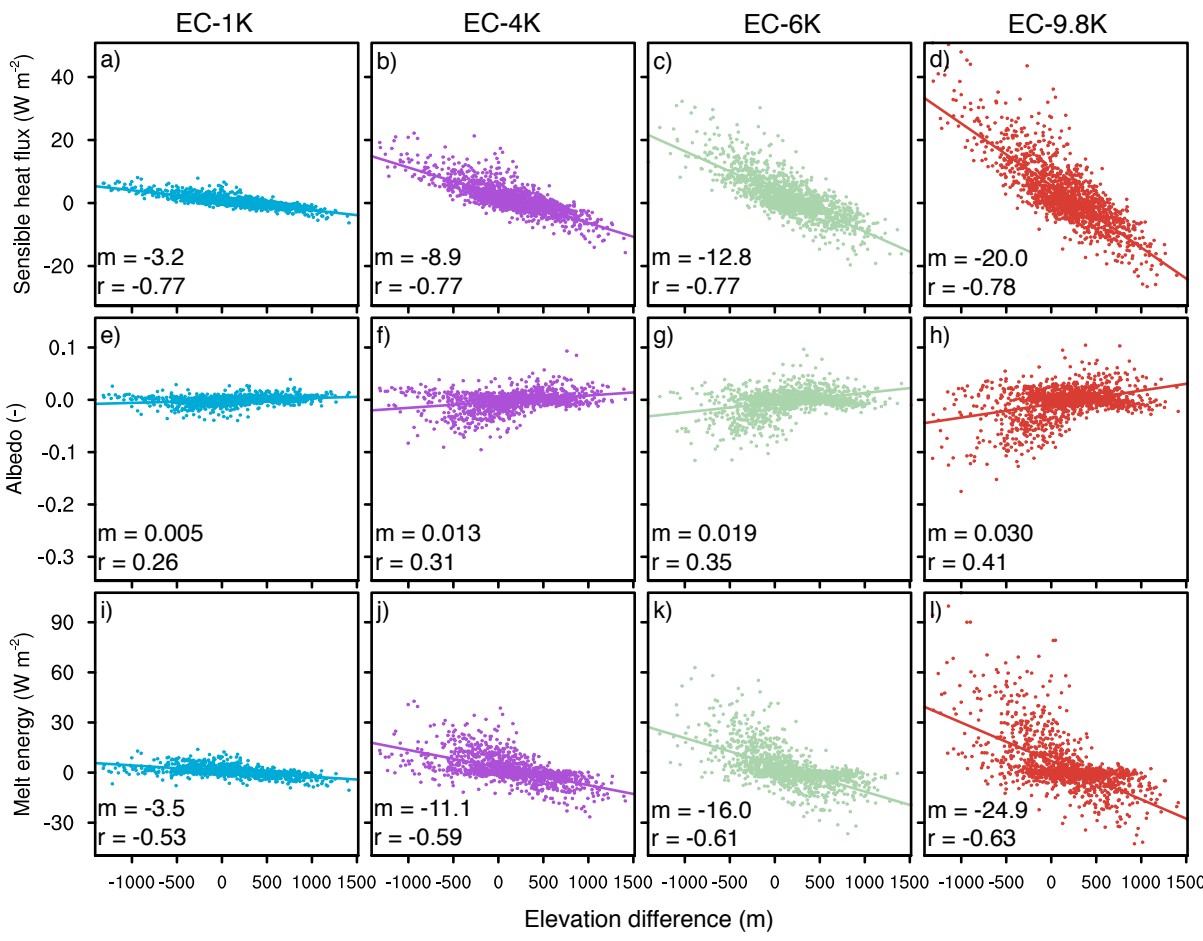

**Figure 3.** Comparison of 1965-2005 summer (JJA) downscaled energy fluxes among four simulations with different elevation corrections for the atmospheric temperature forcing. The first column corresponds to EC-1K, the second to EC-4K, the third to EC-6K and the last to EC-9.8K for a-d) sensible heat flux (W m$^{-2}$), e-h) albedo (-) and i-l) melt energy (W m$^{-2}$). The x- and y-value represent deviation of surface elevation and energy component for each data point with respect to the climate model grid ($\sim$ 100 km) mean. The lines represent least-squares linear regressions. The annotated $m$ is the least-squares linear regression gradient (W m$^{-2}$ km$^{-1}$ or km$^{-1}$ for albedo), $r$ is the correlation coefficient.



**Figure 4.** Climatological (1965-2005) downscaled (5 km) SMB (a, b) and SMB anomalies (c, d) (mm yr$^{-1}$) using lapse rates a) 6 K km$^{-1}$, b) 1 K km$^{-1}$, c) 4 K km$^{-1}$ and d) 9.8 K km$^{-1}$. Anomalies are with respect to the default lapse rate of 6K km$^{-1}$. Solid black contour shows the ice sheet margin. Elevation contours (dashed) are plotted every 500 m. The black line shows the ice sheet margin. Black dots show where differences are significant at the 95% level according to a student t-test.



**Figure 5.** Annual climatology (1965-2005) of EC-1K (left column) and anomalies of EC-6K with respect to EC-1K, which approximate the EC imprint (right column). a,b) near-surface air temperature (K), c,d) turbulent (sensible+latent) heat fluxes (W m$^{-2}$), e,f) net longwave radiation (W m$^{-2}$) and g,h) sea ice concentration (-). Black dots indicate significance at the 95% level according to a students t-test. Positive signs for a-f indicate energy transfer from atmosphere to the surface.





**Table 1.** Gradients ($m$) and correlation with elevation ($r$; unitless) of surface energy and mass balance components as simulated through EC downscaling in CESM1.0. Values correspond to JJA (energy) and annual (mass) averages for the period 1965-2005. Melt energy is the sum of the net shortwave and longwave radiation and the heat fluxes. Surface mass balance is the sum of snowfall and refreezing, minus melt and sublimation.

| | RACMO2.3 | | CESM1.0 | |
| --- | --- | --- | --- | --- |
| | $m$ | $r$ | $m$ | $r$ |
| *Surface energy balance components* | | | | |
| Incoming solar radiation (W m$^{-2}$ km$^{-1}$) | 15.1 | 0.72 | 0.0 | - |
| Albedo (km$^{-1}$) | 0.081 | 0.60 | 0.019 | 0.35 |
| Net solar radiation (W m$^{-2}$ km$^{-1}$) | -19.6 | -0.61 | -3.5 | -0.30 |
| Incoming longwave radiation (W m$^{-2}$ km$^{-1}$) | -17.6 | -0.81 | 0.0 | - |
| Net longwave radiation (W m$^{-2}$ km$^{-1}$) | -3.1 | -0.30 | 8.9 | 0.76 |
| Sensible heat flux (W m$^{-2}$ km$^{-1}$) | -3.8 | -0.42 | -12.8 | -0.77 |
| Latent heat flux (W m$^{-2}$ km$^{-1}$) | -0.2 | -0.02 | -10.8 | -0.76 |
| Ground heat flux (W m$^{-2}$ km$^{-1}$) | 0.4 | 0.05 | 2.1 | 0.46 |
| Melt energy (W m$^{-2}$ km$^{-1}$) | -26.3 | -0.70 | -16.0 | -0.61 |
| *Surface mass balance components* | | | | |
| Snowfall (mm yr$^{-1}$ km$^{-1}$) | -218 | -0.26 | 0 | - |
| Melt (mm yr$^{-1}$ km$^{-1}$) | -717 | -0.70 | -425 | -0.58 |
| Refreezing (mm yr$^{-1}$ km$^{-1}$) | -129 | -0.45 | 62 | 0.49 |
| Sublimation (mm yr$^{-1}$ km$^{-1}$) | 13 | 0.27 | 47 | 0.75 |
| Surface mass balance (mm yr$^{-1}$ km$^{-1}$) | 369 | 0.28 | 439 | 0.58 |



**Table 2.** Simulated whole-ice-sheet SMB, ablation area, total SMB in the ablation and accumulation areas, and prognostic near-surface temperature gradients for the four simulations performed in this study with varying lapse rates, and for the reference regional model RACMO2.3. Values correspond to the climatological (1965-2005) average with the standard deviation in parentheses.

| | RACMO2.3 | EC-1K | EC-4K | EC-6K | EC-9.8K |
|---|---|---|---|---|---|
| Surface mass balance (Gt yr$^{-1}$) | 382 (102) | 326 (122) | 326 (128) | 372 (101) | 367 (125) |
| Ablation area (% of total GrIS area) | 10.9 (2.4) | 16.4 (4.0) | 15.6 (4.1) | 13.4 (3.0) | 13.0 (3.0) |
| SMB in ablation area (Gt yr$^{-1}$) | -138 (45) | -142 (68) | -153 (75) | -128 (50) | -142 (48) |
| SMB in accumulation area (Gt yr$^{-1}$) | 520 (71) | 468 (68) | 480 (78) | 500 (63) | 509 (92) |
| Prognostic temperature lapse rate [JJA] (K km$^{-1}$) | 4.3 (0.2) | 0.5 (0.0) | 2.0 (0.1) | 3.0 (0.1) | 5.0 (0.2) |
| Prognostic temperature lapse rate [DJF] (K km$^{-1}$) | 4.6 (0.2) | 0.8 (0.0) | 2.5 (0.0) | 3.6 (0.0) | 5.8 (0.0) |



**Table 3.** Simulated annual (ANN) and summer (JJA) GrIS averaged components of the surface energy with the standard deviation in parentheses. The period considered is 1965 to 2005. Closest values to RACMO2.3 are given in bold.

|  | RACMO2.3 | | EC-1K | | EC-6K | |
| --- | --- | --- | --- | --- | --- | --- |
|  | ANN | JJA | ANN | JJA | ANN | JJA |
| Albedo (-) |  | 0.81 (0.01) |  | 0.78 (0.02) |  | 0.78 (0.01) |
| Near-surface air temperature (K) | 252.0 (0.8) | 265.6 (0.8) | 252.9 (0.8) | 266.5 (0.9) | **252.0** (0.8) | **265.9** (0.8) |
| Turbulent heat flux (W m$^{-2}$) | 19.1 (0.5) | 2.7 (0.7) | 23.2 (0.8) | **3.0** (1.3) | **21.1** (0.6) | 0.5 (1.0) |
| Net longwave radiation (W m$^{-2}$) | -43.9 (0.9) | -52.4 (2.0) | -45.9 (1.2) | **-47.5** (3.4) | **-43.9** (1.0) | -44.9 (2.7) |
| Melt energy (W m$^{-2}$) | 2.8 (0.5) | 10.2 (1.9) | 3.5 (0.8) | 12.4 (3.1) | **3.2** (0.6) | **11.6** (2.4) |