# Peer review of "Surface mass balance downscaling through elevation classes in an Earth System Model: analysis, evaluation and impacts on the simulated climate"

_The Cryosphere, 2019_

## Referee Comment (RC1) · Anonymous Referee #1 · 22 Jul 2019

General comments

This paper describes an analysis of the online climate downscaling scheme used over the Greenland ice sheet in an older version of CESM, alongside a limited study of how sensitive it is to the temperature lapse rate specified, one of the scheme's key parameters. The topic is timely, although I don't think it's totally clear that this belongs in The Cryosphere rather than Geosci. Model Dev., seeing as it doesn't purport to research anything about the real world, rather evaluate the emergent behaviour of a model parameterisation. That's not meant to imply that I don't think it's an important

subject, and it is valuable to highlight these results to a wider audience than those who work on the development of climate models, as it directly bears on the interpretation of model results that are used widely in the cryospheric community. In general I liked it, and as someone working in this area, I found it practically useful.The paper is well structured and clearly written - my main recommendation for an improvement would simply be to show more figures.

One might suggest a number of improvements to the downscaling scheme itself, of course, but those would be outside the scope of the work actually presented here.

Detail

title: the editor's initial comments have already touched on the title, but I'm not convinced the current title is as clear as it could be. The quantities being actively downscaled are "climate" variables, not the SMB itself.

The authors' replies to this comment from the editor also say they'd prefer to leave "Greenland" out of the title, as the scheme is general. Personally I'd put it in. The other obvious ice sheet application for this is for Antarctica, but circumstances there are rather different. Sub-gridscale variation in SMB components there is more dominated by dynamic weather considerations rather than pure elevation, and temperatures are such that the lower, melt/bare ice albedoes - the only means by which sub-grid variation in shortwave radiation can really enter in this scheme - should play a much smaller role. That being the case, the analysis and component gradients here probably *are* only really applicable to Greenland. Additionally, no comment is made of how the scheme might perform for other ice sheets - perhaps if the authors wanted to leave the title as it is they could include some discussion about how the scheme might be expected to perform on Antarctica, or if applied to paleo ice sheets in other regions?

If they wish to keep the scope to just modern-day Greenland, how about "Downscaling climate through elevation classes for Greenland ice sheet surface mass balance in an ESM: analysis [etc...]"?

page 1, line 5: it would be clearer if you note that RACMO is an RCM

p1,l20: "leading" would be better as "which would lead"

p1,l21: "is losing" would be better as "has lost" if you start the sentence with "Since"

p2,l6: I'd say "ESM" deserved a wider definition than 'a climate model with a carbon cycle'. There are many possible physical components in an ESM, and for certain applications I don't think you would necessarily have to have the carbon cycle part active to still call the model an ESM

p2,l8: does the "SMB" contraction need defining in the Introduction proper rather than just in the abstract, which can sometimes stand alone from the main paper?

p2,l10-25: I didn't think the distinction between methods 2. and 3. was terribly clear, or that the "hybrid" variant used by CESM doesn't really sit within method 2. or 3. The section also suggests it's going to list "state of the art downscaling techniques" in general, but this is a wide field and this list seems far from comprehensive - pattern scaling, EOF methods etc

p3,l1: since CESM1 was superseded by version 2 more than a year ago, I think that somewhere in the introduction it would help if you explicitly noted that you're not using the current release version of CESM, and said why. Perhaps in the Discussion you could also note what, if anything, readers might expect to be different in CESM2, based on what you've learnt and what has changed in the model in the meantime

p4,l24: Why did you use a "minimal", 1K/km lapse rate as a control rather than 0K/km which would effectively deactivate the scheme properly and revert to the type of behaviour seen in most ESMs?

p5,l4: I think it's noted later in the analysis, but you're effectively comparing two (likely completely different) realisations of climate variability during a specific historical period by using an ERA-forced RCM vs the GCM. I think it's worth flagging this up, and anticipating the possible impacts here already.

[Figure]

p5,l9 The framework used from here on does rely rather on fitting simple linear relation-ships to scatter plots of "<variable> vs elevation" from all of Greenland. The apparently wide scatter in the figures often make it look like such a simple relationship really isn't a good way to approach the RACMO data being compared with, although the r values given look higher than the scatter shown in the plot might suggest, so perhaps this is more a presentational issue? Since a universal linear gradient is the paradigm being used in CESM - and the CESM fits do often *look* much more linear - it's not an unjustified way to proceed, but some kind of cautionary note should be put in here that this is a potentially over-simplistic way of looking at regionally heterogeneous data from a much higher resolution study, and that for some variables the scatter makes the fits and gradients reported perhaps more qualitative than quantitative.

p5,l17: I think some 2D plots so that readers can see the regional differences between the RACMO2.3 reference fields you use in this study and your CESM1 SMB would be very useful here. The choice of which figures to put in the main paper in which should be supplementary material will need to be thought about, but in general I think this is the first of a couple of areas in this paper where it would just be useful to be able to see more information than is currently there.

p7,l29: it wasn't immediately obvious to me why the subset of fluxes shown in figure 3 were the "most relevant"

p7,l33: the CESM albedo does not look very sensitive to the lapse rate - probably worth noting that even at the maximum lapse rate you don't even get to half of the RACMO value.

p8,l9: why not actually do the SMB scatter plots and show the gradients? Surely they're important enough to show explicitly?

p8,l31: I really didn't understand the description of the prognostic temperature, or how it was calculated

p9,l9: I still don't understand why you used a 1K/km experiment as the control, rather than 0K/km?

p9,l14: it's not clear in which topography the "mean elevation is lower"

p9,l29: the Supplementary info figure is labelled A1 here, but as S1 in one of the links I was given to download

p9,l30: the large discrepancies in the comparison with the reanalysis may be a place where the fact that the reanalysis and the GCM will have different realisations of internal climate variability really plays a role

p10,l29: as previously noted, I think it's worth flagging up differences between your CESM1 and the new CESM2, which is the version new users will likely pick up. Can you say which, if any, of the recommendations you make have been implemented in the current CESM?

p11,l20: "certain lapse rates score better for some metrics than others" is a little disingenuous, really. You've done a great job of showing that that the components being directly downscaled via the lapse rates generally cannot be made to match the physical elevation gradients for any value of the lapse rate, and that the final SMB you get only scores well because of fortunate cancellation of these significant errors. At this point, the "lapse rate" you specify almost loses a physical meaning - it's no longer a parameter you might desire to constrain directly through observations to match reality, rather a model control you can tune directly to get the final (SMB) result you want without worrying about the fidelity of the underlying components that go into that result. Something along these lines should be noted in this paragraph, I think

p11,l24: implies that it's hard to distinguish between the EC6K and EC9.8K SMB gradients, yet two sentences before states that EC6 has a better SMB gradient but EC9 has the best melt. I'm confused as to whether you can really make a robust distinction between the SMB gradients in the two cases - especially since the SMB vs height

scatter plots are not shown for cases other than EC6. Does the r value on the SMB gradient actually justify distinguishing between the two cases? If, in fact, you're only basing that recommendation on the top line of total GrIS SMB in Table 2, given the size of the standard deviation on the RACMO numbers it would seem difficult to justify saying one is better than the other.

Figures ——-

On the whole I feel that the paper could be usefully improved by tweaking the presentation of the figures. Above I've noted that it would be good if 2d plots of the EC6k vs the RACMO2 reference data could be shown, and the actual SMB scatter plots and fits for EC1, EC4.5, EC6 and EC9.8 rather than only summarising this data in a table. It may be that the authors or editor take a view on which figures belong in the main body of the paper and which in Supplementary information, but I do think it would be ueisul to show them.

Of all the panels of scatter plots, only Figure 3 includes the useful gradient and r values on the scatter plots themselves - it would be useful if Figures 1 and 2 could show this information as well. In Figure 4, why is the absolute value of SMB shown for the EC1K experiment rather than the more useful difference from EC6K, which is how the information for the EC4K and EC9.8K experiments is shown in panels c) and d)?

---

## Referee Comment (RC2) · Anonymous Referee #2 · 23 Jul 2019

**Review of "Surface mass balance downscaling through elevation classes in an Earth System Model: analysis, evaluation and impacts on the simulated climate"**
by R. Sellevold et al.

**Summary:**   The authors present an evaluation of an elevation class scheme applied in the Community Earth System Model 1.0 (CESM1.0). The elevation class scheme allows CESM to simulate surface processes at a sub-grid scale, and allows for interaction between the surface and the atmosphere on the CESM gird scale after surface fluxes are integrated on the CESM grid.  The authors mainly focus on comparing gradients of energy and mass balance components at the sub-grid scale with gradients from the RACMO2.3 RCM, a leading RCM used to simulate surface mass balance (SMB) over the Greenland ice sheet.  CESM captures gradients of SMB effectively as compared with RACMO2.3 but SMB and surface energy balance (SEB) components are not captured as effectively.  Biases in these components tend to compensate for each other, resulting in the effective simulation of SMB gradients.   The authors also find that implementing the elevation class scheme influences the simulation of regional climate around Greenland in CESM.

**General Comments**
I feel that this paper represents an important contribution to our understanding of simulating ice sheet surface mass balance in global climate models.  Implementing such simulation is essential for capturing SMB-climate feedbacks in future climate projections with earth system models, which the authors also briefly address in the study.  The paper is well argued and the analysis and interpretation of results is straightforward and logical.   Sometimes the text becomes a bit wordy and difficult to understand; some suggestions are provided below.  I feel the paper can be excepted with relatively minor revisions. Some general comments:

1. There is a recent study by Alexander et al. (2019) that builds on work of Fischer et al. (2014), which evaluates the impact of elevation classes on simulation of Greenland SMB in the NASA GISS ModelE GCM.   This study was similar in comparing GCM outputs to an RCM, but differs in that the EC simulation was not evaluated at a high resolution as is done here.  These studies should be mentioned and the authors may be interested in exploring similarities or differences in the conclusions at the ESM grid scale.

Alexander, P., LeGrande, A. N., Fischer, E., Tedesco, M., Fettweis, X., Kelley, M., Nowicki, S. M. J., and Schmidt, G. A.: Simulated Greenland ice sheet surface mass balance in the GISS ModelE2 GCM: Role of the ice sheet surface, Journal of Geophysical Research, 124, 750-765, https://doi.org/10.1029/2018JF004772, 2019.

Fischer, R., Nowicki, S., Kelley, M., and Schmidt, G. A.: A system of conservative regridding for ice-atmosphere coupling in a General Circulation Model (GCM), Geoscientific Model Development, 7, 883-907, https://doi.org/10.5194/gmd-7-883-2014, 2014.

2. The authors discuss some gradients for which scatter plots are not included. It would be helpful if the authors could provide additional figures for e.g. downward and upward shortwave and longwave radiation, snow accumulation and refreezing. These figures could be included as additional panels in Figures 1 and 2 or supplemental figures at the authors' discretion.
3. It is not clear in the text that the SEB terms are computed for JJA, while SMB terms are computed annually. The authors should make this difference clear in the methods and results sections.

**Specific Comments**
1. Figure 1: Though not essential, it would be helpful if the authors add text or a legend on one of the figures to show that black is RACMO2.3 and blue is CESM1.0. Also it would be helpful if the authors specify the sign convention (+ down) in the legend and text. Mention y-axis scale differences in the legend. Also, what is meant by "several summer SEB components". Is only a subset of years used to calculate the gradients? If so this should be made clear in the text.
2. Figure 2: Again, include a legend on the figure if possible. Mention difference in y-axis scales in the legend.
3. Table 2: Are the standard deviation values annual values? Please specify.
4. P. 1, Line 5: Change "from RACMO2.3" to "from the RACMO2.3 regional climate model." The model has not been introduced yet.
5. P. 1, Lines 11-12: The topographic smoothing affects the atmospheric simulation, while the elevation class technique cools the surface by another means. It seems the technique doesn't really "correct" the bias, but rather "compensates" for it by correcting a bias associated with the coarse ESM resolution. Is this the case? Please clarify here.
6. P. 2, Lines 24-25: Here the authors might mention that a benefit of the "online" approach is that it is able to capture feedbacks between the downscaled surface simulation and the atmospheric component of the ESM.
7. P. 2, Lines 26-30: As noted above, an elevation class scheme has also been implemented in the NASA GISS ModelE GCM in an "online" manner as discussed by Fischer et al. (2014) and evaluated on the ModelE grid by Alexander et al. (2019). However, I believe the authors are correct that the effects of downscaling on the finer resolution grid representation of SMB and SEB has not been evaluated in detail. These studies should be mentioned and the authors should make clear the distinction between evaluation on the coarse resolution model grid, and at the finer scale.
8. P. 3, Line 19: I believe the authors are referring to downscaling using elevation classes, but this is not entirely clear. Perhaps the sentence can be revised to read something like "A static ice sheet surface that corresponds to present-day observations (Bamber et al., 2013) is used to downscale SMB and other quantities through the elevation class scheme."
9. P. 3, Line 21 – P. 4, Line 21: This section is a bit confusing. The steps in the elevation class scheme are not entirely clear. I believe the steps are as follows: 1. A set of elevation classes is defined and for each grid cell containing ice. 2. Some atmospheric

quantities are downscaled by elevation.  2.  The surface model is run for each elevation class, forced with the downscaled quantities.  3.  Surface model outputs are averaged to the ESM grid, weighting for the percentage of each elevation class within each ESM grid cell, and these integrated quantities feed back to the atmosphere.   Perhaps these steps can be clarified in the paragraph on P. 3, lines 21-24, and this will make the following material clear, or the text can be revised to mention one step at a time.

10. P. 3, Line 27: How is the weight of each elevation class within a grid cell determined?

11. P. 3, Line 28: I believe "average" is referring to the average surface to atmosphere fluxes, and outputs such as SMB and SMB components, but this is not clear.

12. P. 3, Line 33:  Clarify to read "all ECs within a grid cell."

13. P. 4, Lines 2-8: It would be helpful if the authors can reiterate here which terms are common to all ECs within a grid cell, and which terms vary by EC as a result of downscaling.   In particular it should be mentioned that albedo is calculated interactively within the model for each EC based on snow properties / snow depth over ice.

14. P. 4, Lines 11-12:  Note that these quantities are calculated on the ESM grid.  Also, aren't these quantities calculated by the atmospheric model CAM4 and not CLM4?

15. P. 4, Line 30:  Although the details of the setup are described by Vizcaino et al., the authors should mention briefly what forcing is applied (e.g. sea ice/ocean temperatures/ atmospheric nudging).

16. P. 5, Line 15:  Make clear why it is necessary to subtract the average CESM grid value from the RACMO2.3 grid cell values.  I think this is to only capture gradients within grid cells, and not at the coarser resolution.

17. P. 5, Line 19:  "mean elevation" is confusing.  Perhaps use "on the CESM grid".

18. P. 5, Line 23:  "..comparison of the downscaled SEB components via EC and RCM"  is confusing.  It is the gradients that are being compared.  Revise to something like "…comparison of SEB component gradients for CESM1.0 ECs and the RACMO2.3 RCM."

19. P. 6, Line 10:  Change "opposite gradient" to "opposite elevation gradient" for clarity.

20. P. 6, Lines 16-18:  The difference in sign here makes this a bit confusing.  Including the longwave components in Fig. 1 or in a supplemental figure would help the reader to easily visualize this.

21. P. 6, Lines 22-23:  Suggest changing "null gradients of incoming radiation in the model and weaker albedo gradients" to "a null gradient of incoming radiation in CESM1.0 and weaker albedo gradients than in RACMO2.3, leading to a smaller gradient in net shortwave radiation."  Also, I believe this sentence is only referring to shortwave radiation, but this is not mentioned.  Please clarify.

22. P. 7, Lines 11-12:  What is the value of the gradient for CESM1.0?

23. P. 7, Lines 14-15: Again, it would be interesting to see the figures for snowfall and refreezing.

24. P. 8, Lines 1-3:  Not sure what is meant by "non-null variations".  It would be clearer to simply note that the albedo gradient increases with increasing lapse rate, as shown in Figure 3.

25. P. 8, Lines 14-15:  Any idea why there is a reversal for the 9.8K/km case?

26. P. 8, Line 19:  This is the first time interannual variability is mentioned. Perhaps introduce this with a separate sentence, explaining why interannual variability is interesting in this case and not elsewhere in the study.
27.  P. 9, Lines 13-15:  This is confusing and should be clarified. I think the authors mean that the mean grid cell elevation is lower than the elevation of the ice sheet, so without ECs, the simulated ice sheet is higher in elevation.  This effect was also observed by Alexander et al. (2019).
28. P. 9, Line 27:  Any idea as to why incoming longwave radiation changes?
29. P. 9, Lines 29-34: It seems ERA-Interim is used for locations outside of the RACMO2.3 domain?  Please make this clear.  Also specify here which fields are compared.
30. P. 9, Line 33: change "as with RACMO2.3" to "as differences with RACMO2.3" for clarity.
31. P. 10, Lines 4-5:  Other studies (e.g. Vizcaino et al., 2013; Alexander et al., 2019) have evaluated the EC method at the coarse resolution but not at a higher resolution as done here. This should be clarified here.
32. P. 10, Line 6:  "Linear fits" of what?  Please clarify.
33. P. 10, Line 22:  Change "enables to explore the interaction with" to "enables exploration of the interaction between the high-resolution surface simulation and…"
34. P. 10, Line 25: Change "to RCM" to "to the RACMO2.3 RCM".
35. P. 11, Line 25:  It could be that improving representation of physical processes at the elevation class scale will allow for a better identification of the optimal lapse rate.  This could be mentioned here, if the authors agree.
36. P. 11, Line 32:  Clarify "for radiation", e.g. "for apparent biases in gradients of net radiation"
37. P. 12, Line 11: Clarify "more adequate".

**Technical Corrections**
1. P. 1, Line 2: Remove "the" before "surface mass balance (SMB) modeling"
2. P. 1, Line 4: Change "elevation dependent" to "elevation-dependent"
3. P. 1, Line 20:  Change "leading" to "which would lead".
4. P. 1, Line 21: Change "is losing mass" to "has been losing mass".
5. P. 2, Line 9:  Perhaps change "seem required" to "are likely required"?
6. P. 2, Line 12: The van Kampenhout paper year can be changed to 2019, with the reference to the final revised paper updated in the reference list
7. P. 2, Line 17: Revise "Statistical downscaling, which uses elevation corrections on…" to "Statistical downscaling uses elevation corrections to…"
8. P. 5, Line 23:  Change "r-value" to "r-values".
9. P. 5, Line 25:  Change "…solar radiation is not downscaled so that all ECs receive…" to "…solar radiation is not downscaled.  As a result, all ECs within a grid cell receive…"
10. P. 6, Line 4:  Change "more correlated" to "better correlated".
11. P. 6, Lines 20-21:  Change "The net radiation gradients…" to "The net radiation gradient in CESM1.0 is 5.4 W m$^{-2}$ km$^{-1}$ and in RACMO2.3 is -22.6 W m$^{-2}$ km$^{-1}$ (Table 1)."
12. P. 7, Line 5: Change "low ECs" to "low elevation ECs"

13. P. 7, Line 24: Change "compensates the biases" to "compensates for the biases"
14. P. 7, Line 26: Change "compensates this" to "compensates for this"
15. P. 8, Line 27: Change "therefore not only reflecting" to "therefore reflect more than just"
16. P. 8, Line 30:  Change "high ECs" to "high elevation ECs" and "low ECs" to "low elevation ECs"
17. P. 8, Line 34:  Change "lower than the magnitude of the respective"  to "lower in magnitude than the respective"
18. P. 8, Line 35: Change "gradient is less" to "gradient is also less"
19. P. 9, Line 27: Change "overcompensated by" to "overcompensated for by"
20. P. 10, Lines 11-13: Suggest revising the sentence to read:  "However, one of the limitations of comparing with an RCM is that unlike an ESM, the RCM is laterally forced with reanalysis.  Also, there are fundamental differences in the physical schemes and simulated climate components between the ESM and RCM compared here."
21. P. 10, Lines 13-14:  Change "net longwave" to "net longwave radiation"
22. P. 11, Line 1:  Change "although it varies" to "despite the fact that it varies"
23. P. 12, Lines 2-3:  Change "efficient to generate" to "efficiently generates"

---

## Referee Comment (RC3) · Anonymous Referee #3 · 25 Jul 2019

General synopsis This is a useful contribution about the novel application of using Earth System Models (ESM) with downscaling, via sub-gridcell elevation classes, to simulate Greenland Ice Sheet surface mass balance. Although it is fairly model-specific (based on the CESM1.0 ESM), this paper should be of broad interest to the GrIS SMB modelling and Greenland climate communities.

Some previous highly relevant literature is missing or can be better acknowledged (see comments below).

I wonder whether the CESM results can be compared with MAR as well as RACMO, for an independent RCM model check (and since MAR is the main alternative RCM currently used for Greenland)?

Specific comments

Page 1, lines 15-16 re. strong Arctic warming: Please add the following reference to those cited: Overland, J.E. and Hanna, Edward and Hanssen-Bauer, I. and Kim, S.-J. and Walsh, J.E. and Wang, M. and Bhatt, U.S. and Thoman, R.L. (2018) Surface air temperature. Arctic Report Card , NOAA. https://www.arctic.noaa.gov/Report-Card/Report-Card-2018/ArtMID/7878/ArticleID/783/Surface-Air-Temperature

P1, L21 re. "GrIS is losing mass at an accelerated rate": please add the following highly relevant references to those cited: Edward Hanna, Francisco J Navarro, Frank Pattyn, Catia M Domingues, Xavier Fettweis, Erik R Ivins, Robert J Nicholls, Catherine Ritz, Ben Smith, Slawek Tulaczyk, Pippa L Whitehouse, H Jay Zwally (2013) Ice sheet mass balance and climate change. Nature 498, 51-59. Bamber, JL et al. (2018): The land ice contribution to sea level during the satellite era. Environmental Research Letters, 13(6), 063008,

P2, L8: should also add there is a significant disparity between different model estimates of GrIS SMB (Fettweis 2018): Fettweis, X. (2018) The SMB Model Intercomparison (SMBMIP) over Greenland: first rlts. AGU Fall Meeting talk archived at: https://orbi.uliege.be/handle/2268/232923.

P2, L19: re. statistical downscaling please add the following highly relevant references: Hanna et al. (2011) AND Wilton et al. (2017) DJ Wilton, A Jowett, E Hanna, GR Bigg, MR Van Den Broeke, X Fettweis, ...(2017) High resolution (1 km) positive degree-day modelling of Greenland ice sheet surface mass balance, 1870–2012 using reanalysis data. Journal of Glaciology 63 (237), 176-193

E Hanna, P Huybrechts, J Cappelen, K Steffen, RC Bales, E Burgess, ...(2011) Green-

land Ice Sheet surface mass balance 1870 to 2010 based on Twentieth Century Re-analysis, and links with global climate forcing. Journal of Geophysical Research: Atmospheres 116 (D24)

P2, L32: While the motivation for the study is good as stated, can you make it clear in this sentence/paragraph whether you investigated precipitation downscaling as well as temperature downscaling?

P3, L26: How was this number of elevation classes chosen? Would having a greater number of classes improve the results?

P3, L34 "Incoming radiation, precipitation and wind are kept constant across all ECs" - Is this a potential limitation of this study or could improvements be made here?

P5, L8 "snow when near-surface temperatures are between -7˚oC and -1˚oC" – the latter value (-1C) seems quite a low upper threshold for snow?

P10, LL4-5 "the first time the EC method for downscaling from a global climate model of ∼100 km to the much higher resolution (5 km) of an ice sheet model" – point out that this kind and magnitude of statistical downscaling has been previously successfully used in downscaling meteorological reanalysis data from ∼100-km resolution to 5-km resolution (Hanna et al. 2005 & 2011, Wilton et al. 2017). E Hanna, P Huybrechts, I Janssens, J Cappelen, K Steffen, A Stephens (2005) Runoff and mass balance of the Greenland ice sheet: 1958–2003. Journal of Geophysical Research: Atmospheres 110 (D13) (Other two references details are above.)

P11, LL9-11: The recommended implementation of a precipitation phase downscaling scheme doesn't really solve the great challenge of overall elevation correction for precipitation. This paragraph therefore sounds a little weak as currently stated – can the authors strengthen their argument here?

P12, L4 "Our sensitivity experiments reveal that a larger lapse rate for the temperature correction results in higher melt energy gradients" – isn't this rather an obvious and

unsurprising result? – perhaps rephrase?

---

## Author Response (AR1)

Reviewers comments in black
*Author's response in red*

**Summary of major changes**

**Figure 1 and 2.** The *m* and *r* have been added to each panel, in addition RACMO2.3 and CESM1.0 in text are annotated to the first figure in colors corresponding to their scatter and lines. Y-axis is changed to show the same range for the same units.

**Figure 2.** Added an additional panel containing snowfall.

**Figure 3.** Added an additional row with SMB vs elevation for each of the lapse rates.

**Figure 4**. Added a spatial map of the RACMO2.3 reference data. Changed panel (b) to show EC-1K minus EC-6K.

**Supplementary figure.** A supplementary figure similar to Figure 1 has been added showing the incoming and outgoing solar and longwave components.

[Figure]

Figure 1

[Figure]

Figure 2

[Figure]

Figure 3

[Figure]

Figure 4

[Figure]

Figure S1

**Reviewer #1:**

**General comments**

This paper describes an analysis of the online climate downscaling scheme used over the Greenland ice sheet in an older version of CESM, alongside a limited study of how sensitive it is to the temperature lapse rate specified, one of the scheme's key parameters. The topic is

timely, although I don't think it's totally clear that this belongs in The Cryosphere rather than Geosci. Model Dev., seeing as it doesn't purport to research anything about the real world, rather evaluate the emergent behaviour of a model parameterisation. That's not meant to imply that I don't think it's an important subject, and it is valuable to highlight these results to a wider audience than those who work on the development of climate models, as it directly bears on the interpretation of model results that are used widely in the cryospheric community. In general I liked it, and as someone working in this area, I found it practically useful.The paper is well structured and clearly written - my main recommendation for an improvement would simply be to show more figures.

*We thank the reviewer for this feedback, and we will expand on the figures in the manuscript, in addition to adding more figures to the supplementary material. We submitted this manuscript to TCD instead of GMD, as the study is more focused on physical analysis of surface fluxes and climate impacts resulting from the method.*

One might suggest a number of improvements to the downscaling scheme itself, of course, but those would be outside the scope of the work actually presented here.

**Detail**

title: the editor's initial comments have already touched on the title, but I'm not convinced the current title is as clear as it could be. The quantities being actively downscaled are "climate" variables, not the SMB itself.

The authors' replies to this comment from the editor also say they'd prefer to leave "Greenland" out of the title, as the scheme is general. Personally I'd put it in. The other obvious ice sheet application for this is for Antarctica, but circumstances there are rather different. Sub-gridscale variation in SMB components there is more dominated by dynamic weather considerations rather than pure elevation, and temperatures are such that the lower, melt/bare ice albedoes - the only means by which sub-grid variation in shortwave radiation can really enter in this scheme - should play a much smaller role. That being the case, the analysis and component gradients here probably *are* only really applicable to Greenland. Additionally, no comment is made of how the scheme might perform for other ice sheets - perhaps if the authors wanted to leave the title as it is they could include some discussion about how the scheme might be expected to perform on Antarctica, or if applied to paleo ice sheets in other regions?

If they wish to keep the scope to just modern-day Greenland, how about "Downscaling climate through elevation classes for Greenland ice sheet surface mass balance in an ESM: analysis [etc...]"?

*We consider we are downscaling the SMB and surface energy fluxes and not the climate, as the downscaling is done in the land component of the CESM as opposed to e.g., grid refinement in the atmospheric component or statistical downscaling of climate variables.*

*We are aware of the current application of the EC method to some other ice sheets, but we prefer not to speculate about how adequate it would be for those. Instead, we think future work can use our study as to guide the evaluation to other (paleo) ice sheets.*

*The title of the manuscript is changed to "Surface mass balance downscaling through elevation classes in an Earth System Model: application to the Greenland ice sheet".*

page 1, line 5: it would be clearer if you note that RACMO is an RCM

*Changed to "from the regional climate model RACMO2.3"*

p1,l20: "leading" would be better as "which would lead"

*Changed accordingly*

p1,l21: "is losing" would be better as "has lost" if you start the sentence with "Since"

*Changed accordingly*

p2,l6: I'd say "ESM" deserved a wider definition than 'a climate model with a carbon cycle'. There are many possible physical components in an ESM, and for certain applications I don't think you would necessarily have to have the carbon cycle part active to still call the model an ESM

*Changed to: "... (ESMs; coupled climate models capable of simulating the Earth's chemical and biological processes, in addition to the physical processes)".*

p2,l8: does the "SMB" contraction need defining in the Introduction proper rather than just in the abstract, which can sometimes stand alone from the main paper?

*SMB is defined in the introduction now (p.2, l.7)*

p2,l10-25: I didn't think the distinction between methods 2. and 3. was terribly clear, or that the "hybrid" variant used by CESM doesn't really sit within method 2. or 3. The section also suggests it's going to list "state of the art downscaling techniques" in general, but this is a wide field and this list seems far from comprehensive - pattern scaling, EOF methods etc

*This is a good point, and it will be too much to go through all of the state of the art downscaling methods. Therefore it is changed to "Most common downscaling techniques for the GrIS SMB are".*

*The distinction between method 2 and 3 is that method 2 only allows for applying statistical corrections to output of a model. This is, as you mention, a very large field containing EOF methods and pattern scaling, in addition to the rapidly growing toolbox of machine learning techniques. However, for a method to qualify for the hybrid approach the SEB/SMB needs to be explicitly calculated after some statistical downscaling of the atmospheric variables involved.*

p3,l1: since CESM1 was superseded by version 2 more than a year ago, I think that somewhere in the introduction it would help if you explicitly noted that you're not using the current release version of CESM, and said why. Perhaps in the Discussion you could also note what, if anything, readers might expect to be different in CESM2, based on what you've learnt and what has changed in the model in the meantime

*This study was based on CESM1.0 as the elevation classes were first introduced in this model version. We are preparing a short follow-up study using CESM2. CESM2 uses some of the recommendations made in the text (e.g., a lapse rate for incoming longwave radiation, a lower ice-albedo, and precipitation phase corrections based on surface temperatures).*

p4,l24: Why did you use a "minimal", 1K/km lapse rate as a control rather than 0K/km which would effectively deactivate the scheme properly and revert to the type of behaviour seen in most ESMs?

*We feared that using a lapse rate of 0 K km$^{-1}$ could have unwanted consequences: the model was designed to have activate elevation class with a certain lapse rate, so completely disabling this feature would potentially lead to model artefacts.*

p5,l4: I think it's noted later in the analysis, but you're effectively comparing two (likely completely different) realisations of climate variability during a specific historical period by using an ERA-forced RCM vs the GCM. I think it's worth flagging this up, and anticipating the possible impacts here already.

*Have added: "As we are only comparing CESM1.0 simulations with identical initial conditions, we are likely to sample a different realization of climate variability than the reanalysis forced RACMO2.3." in section 2.3.*

p5,l9 The framework used from here on does rely rather on fitting simple linear relationships to scatter plots of " vs elevation" from all of Greenland. The apparently wide scatter in the figures often make it look like such a simple relationship really isn't a good way to approach the RACMO data being compared with, although the r values given look higher than the scatter shown in the plot might suggest, so perhaps this is more a presentational issue? Since a universal linear gradient is the paradigm being used in CESM - and the CESM fits do often *look* much more linear - it's not an unjustified way to proceed, but some kind of cautionary note should be put in here that this is a potentially over-simplistic way of looking at regionally

heterogeneous data from a much higher resolution study, and that for some variables the scatter makes the fits and gradients reported perhaps more qualitative than quantitative.

*This is a very valid point. We believe the scatter itself might appear as larger than it statistically is, especially for RACMO, and the reason for the r-values to look higher than the scatter shown is due to the very high number of points in this plot (13,311 for RACMO and 1,551 for CESM as stated in p5,l17). Also, the temperature forcing itself is a linear regression onto the elevation, which makes seeing how (non-)linear different quantities respond to such a forcing an interesting point in itself through the linear correlation coefficient with elevation.*

p5,l17: I think some 2D plots so that readers can see the regional differences between the RACMO2.3 reference fields you use in this study and your CESM1 SMB would be very useful here. The choice of which figures to put in the main paper in which should be supplementary material will need to be thought about, but in general I think this is the first of a couple of areas in this paper where it would just be useful to be able to see more information than is currently there.

*We have added a map of the RACMO2.3 data to Fig. 4, and made EC-1K to an anomaly map wrt. EC-6K.*

p7,l29: it wasn't immediately obvious to me why the subset of fluxes shown in figure 3 were the "most relevant"

*It is because we focus on the fluxes that are controlling the downscaling: the turbulent fluxes (sensible), net shortwave (albedo), and melt energy. We will also add a fourth row with SMB in the final version, as this might be the most relevant of all to show here.*

p7,l33: the CESM albedo does not look very sensitive to the lapse rate - probably worth noting that even at the maximum lapse rate you don't even get to half of the RACMO value.

*The numerical difference in albedo gradient is not very large indeed. However, we feel that it is still sensitive to the forcing lapse rate, as even a small change in the albedo gradient leads to a much larger change in $SW_{net}$ gradients.*

*Added: "Even with the maximum lapse rate forcing, CESM1.0 is only able to produce an albedo gradient that is 35 % of the RACMO2.3 gradient."*

p8,l9: why not actually do the SMB scatter plots and show the gradients? Surely they're important enough to show explicitly?

*Fig. 3 will be updated to also show the SMB scatter plots.*

p8,l31: I really didn't understand the description of the prognostic temperature, or how it was calculated

*The prognostic temperature is calculated at each EC in CLM as a result of the calculated energy fluxes and exchanges. Therefore, it is different from the forcing temperature.*

p9,l9: I still don't understand why you used a 1K/km experiment as the control, rather than 0K/km?

*Please see answer before.*

p9,l14: it's not clear in which topography the "mean elevation is lower"

*In this part, we mean that the atmospheric grid cell elevation is lower than the land grid cell elevation, even though both are on a 1° grid cell as the atmospheric model requires a more smoothed topography to not force the highest wavenumbers that can cause noise. We have changed the sentence to: "First, because the atmospheric topography is more smoothed than the topography in the ice-sheet covered land grid cell, the atmospheric mean elevation per grid cell is lower than the land model mean elevation per grid cell."*

p9,l29: the Supplementary info figure is labelled A1 here, but as S1 in one of the links I was given to download

*It appears as S1 in my PDF. It was labelled A1 during the initial submission, but was corrected and now appears as S1 in the public discussion paper.*

p9,l30: the large discrepancies in the comparison with the reanalysis may be a place where the fact that the reanalysis and the GCM will have different realisations of internal climate variability really plays a role

*We agree. We have added: "As we are only comparing CESM1.0 simulations with identical initial conditions, we are likely to sample a different realization of climate variability than the reanalysis forced RACMO2.3" to section 2.3 so this difference is already expected earlier.*

*We have also changed this part to: "However, the differences between EC-1K and EC-6K are small compared to the difference between these simulations and ERA-Interim, likely due to different realizations of internal climate variability. This precludes a robust conclusion."*

p10,l29: as previously noted, I think it's worth flagging up differences between your CESM1 and the new CESM2, which is the version new users will likely pick up. Can you say which, if any, of the recommendations you make have been implemented in the current CESM?

*Of the recommendations made in this paper, the following were implemented in CESM2:*

- *Lowered albedo*
- *Downscaling of incoming longwave radiation with a fixed elevation dependent gradient*
- *Downscaling of precipitation phase based on surface temperature*
- *More advanced firn simulation*

*We decided not to include references to CESM2, both to avoid confusing reader and as the effects of these different parameterizations have on the downscaled SMB is not yet documented.*

p11,l20: "certain lapse rates score better for some metrics than others" is a little disingenuous, really. You've done a great job of showing that that the components being directly downscaled via the lapse rates generally cannot be made to match the physical elevation gradients for any value of the lapse rate, and that the final SMB you get only scores well because of fortunate cancellation of these significant errors. At this point, the "lapse rate" you specify almost loses a physical meaning - it's no longer a parameter you might desire to constrain directly through observations to match reality, rather a model control you can tune directly to get the final (SMB) result you want without worrying about the fidelity of the underlying components that go into that result. Something along these lines should be noted in this paragraph, I think

*Yes, we agree in that the lapse rate is the tuning control to redistribute energy within a grid cell. We have removed the statement: "certain lapse rates score better …".*

p11,l24: implies that it's hard to distinguish between the EC6K and EC9.8K SMB gradients, yet two sentences before states that EC6 has a better SMB gradient but EC9 has the best melt. I'm confused as to whether you can really make a robust distinction between the SMB gradients in the two cases - especially since the SMB vs height scatter plots are not shown for cases other than EC6. Does the r value on the SMB gradient actually justify distinguishing between the two cases? If, in fact, you're only basing that recommendation on the top line of total GrIS SMB in Table 2, given the size of the standard deviation on the RACMO numbers it would seem difficult to justify saying one is better than the other.

*We will add an additional row to Fig. 3 with SMB gradients which clearly show that the SMB gradient in EC-9.8K is very steep compared to RACMO.*

**Figures**

On the whole I feel that the paper could be usefully improved by tweaking the presentation of the figures. Above I've noted that it would be good if 2d plots of the EC6k vs the RACMO2 reference data could be shown, and the actual SMB scatter plots and fits for EC1, EC4.5, EC6 and EC9.8 rather than only summarising this data in a table. It may be that the authors or editor take a view on which figures belong in the main body of the paper and which in Supplementary information, but I do think it would be uesful to show them.

*We have followed the recommendations of the reviewer and will add the SMB scatterplots to Fig. 3.*

Of all the panels of scatter plots, only Figure 3 includes the useful gradient and r values on the scatter plots themselves - it would be useful if Figures 1 and 2 could show this information as well. In Figure 4, why is the absolute value of SMB shown for the EC1K experiment rather than the more useful difference from EC6K, which is how the information for the EC4K and EC9.8K experiments is shown in panels c) and d)?

*We follow the reviewers recommendation of adding the m and r values directly to the plots.*

**Reviewer #2:**

**Summary**: The authors present an evaluation of an elevation class scheme applied in the Community Earth System Model 1.0 (CESM1.0). The elevation class scheme allows CESM to simulate surface processes at a sub-grid scale, and allows for interaction between the surface and the atmosphere on the CESM gird scale after surface fluxes are integrated on the CESM grid. The authors mainly focus on comparing gradients of energy and mass balance components at the sub-grid scale with gradients from the RACMO2.3 RCM, a leading RCM used to simulate surface mass balance (SMB) over the Greenland ice sheet. CESM captures gradients of SMB effectively as compared with RACMO2.3 but SMB and surface energy balance (SEB) components are not captured as effectively. Biases in these components tend to compensate for each other, resulting in the effective simulation of SMB gradients. The authors also find that implementing the elevation class scheme influences the simulation of regional climate around Greenland in CESM.

**General Comments**

I feel that this paper represents an important contribution to our understanding of simulating ice sheet surface mass balance in global climate models. Implementing such simulation is essential for capturing SMB-climate feedbacks in future climate projections with earth system models, which the authors also briefly address in the study. The paper is well argued and the analysis and interpretation of results is straightforward and logical. Sometimes the text becomes a bit wordy and difficult to understand; some suggestions are provided below. I feel the paper can be excepted with relatively minor revisions. Some general comments:

*Thank you for your positive feedback, we have addressed individual comments below.*

1. There is a recent study by Alexander et al. (2019) that builds on work of Fischer et al. (2014), which evaluates the impact of elevation classes on simulation of Greenland SMB in the NASA GISS ModelE GCM. This study was similar in comparing GCM outputs to an RCM, but differs in that the EC simulation was not evaluated at a high resolution as is done here. These studies should be mentioned and the authors may be interested in exploring similarities or differences in the conclusions at the ESM grid scale.

Alexander, P., LeGrande, A. N., Fischer, E., Tedesco, M., Fettweis, X., Kelley, M., Nowicki, S. M. J., and Schmidt, G. A.: Simulated Greenland ice sheet surface mass balance in the GISS ModelE2 GCM: Role of the ice sheet surface, Journal of Geophysical Research, 124, 750-765, https://doi.org/10.1029/2018JF004772, 2019.

Fischer, R., Nowicki, S., Kelley, M., and Schmidt, G. A.: A system of conservative regridding for ice-atmosphere coupling in a General Circulation Model (GCM), Geoscientific Model Development, 7, 883-907, https://doi.org/10.5194/gmd-7-883-2014, 2014.

*We include references to these papers in the revised manuscript.*

2. The authors discuss some gradients for which scatter plots are not included. It would be helpful if the authors could provide additional figures for e.g. downward and upward shortwave and longwave radiation, snow accumulation and refreezing. These figures could be included as additional panels in Figures 1 and 2 or supplemental figures at the authors' discretion.

*We agree that these figures would provide the readers with additional insight, and will add them to the supplementary material.*

3. It is not clear in the text that the SEB terms are computed for JJA, while SMB terms are computed annually. The authors should make this difference clear in the methods and results sections.

*In the model, both the SEB and SMB terms are computed every 30 minutes. However, our analysis focuses on JJA for SEB terms (as melt on the ice sheet is generated in summer) and annual SMB. As we see this is a point of confusion, we have changed in the description of the SEB calculation: "At each EC, an energy balance model is used to calculate the surface energy balance every 30 minutes" and: "SMB (...) is calculated at each EC, with the same frequency as the SEB calculation, as".*

**Specific Comments**

1. Figure 1: Though not essential, it would be helpful if the authors add text or a legend on one of the figures to show that black is RACMO2.3 and blue is CESM1.0. Also it would be helpful if the authors specify the sign convention (+ down) in the legend and text. Mention y-axis scale

differences in the legend. Also, what is meant by "several summer SEB components". Is only a subset of years used to calculate the gradients? If so this should be made clear in the text.

*All SEB components plotted are the multi-year means from the full simulation (1965-2005). Several refers to showing more than one SEB component. Rather than mentioning the y-axis scale, we will update the figures to use the same y-axis where quantities are of the same unit. We will also include annotated text on the plot to show which is RACMO and which is CESM.*

2. Figure 2: Again, include a legend on the figure if possible. Mention difference in y-axis scales in the legend.

*This figure is updated with a legend, and the y-axis are changed to be the same for every panel.*

3. Table 2: Are the standard deviation values annual values? Please specify.

*Yes, the standard deviation are annual values, except for the prognostic temperatures where the standard deviations are JJA and DJF.*

4. P. 1, Line 5: Change "from RACMO2.3" to "from the RACMO2.3 regional climate model." The model has not been introduced yet.

*We will change accordingly.*

5. P. 1, Lines 11-12: The topographic smoothing affects the atmospheric simulation, while the elevation class technique cools the surface by another means. It seems the technique doesn't really "correct" the bias, but rather "compensates" for it by correcting a bias associated with the coarse ESM resolution. Is this the case? Please clarify here.

*Yes, compensates is a more appropriate word. Changed "corrects" to "compensates for" at p.1, l.11 and p.11, l26.*

6. P. 2, Lines 24-25: Here the authors might mention that a benefit of the "online" approach is that it is able to capture feedbacks between the downscaled surface simulation and the atmospheric component of the ESM.

*Added: "A benefit of this "online" approach is that it is able to capture feedbacks between the downscaled surface simulation and the atmospheric component of the ESM."*

7. P. 2, Lines 26-30: As noted above, an elevation class scheme has also been implemented in the NASA GISS ModelE GCM in an "online" manner as discussed by Fischer et al. (2014) and evaluated on the ModelE grid by Alexander et al. (2019). However, I believe the authors are correct that the effects of downscaling on the finer resolution grid representation of SMB and SEB has not been evaluated in detail. These studies should be mentioned and the authors

should make clear the distinction between evaluation on the coarse resolution model grid, and at the finer scale.

*This is clearly relevant literature that should be reviewed in this section, so we will add references to the articles.*

8. P. 3, Line 19: I believe the authors are referring to downscaling using elevation classes, but this is not entirely clear. Perhaps the sentence can be revised to read something like "A static ice sheet surface that corresponds to present-day observations (Bamber et al., 2013) is used to downscale SMB and other quantities through the elevation class scheme."

*A modified version of this sentence is added: " A static ice sheet surface that corresponds to present-day observations (Bamber et al., 2013) is used to downscale SMB, energy fluxes and other quantities at the land/atmosphere interface through the EC scheme."*

9. P. 3, Line 21 – P. 4, Line 21: This section is a bit confusing. The steps in the elevation class scheme are not entirely clear. I believe the steps are as follows: 1. A set of elevation classes is defined and for each grid cell containing ice. 2. Some atmospheric quantities are downscaled by elevation. 2. The surface model is run for each elevation class, forced with the downscaled quantities. 3. Surface model outputs are averaged to the ESM grid, weighting for the percentage of each elevation class within each ESM grid cell, and these integrated quantities feed back to the atmosphere. Perhaps these steps can be clarified in the paragraph on P. 3, lines 21-24, and this will make the following material clear, or the text can be revised to mention one step at a time.

*We follow the reviewers suggestions, and have added the steps:*
*"The steps for of the EC calculation in an ESM are roughly as follows*

*    1. A set of elevation classes are defined for each (partially) glaciated grid cell in the land model*
*    2. A selected set of atmospheric variables are downscaled by applying simple elevation corrections (typically, prescribed lapse rates)*
*    3. The land model calculates the SEB and SMB per EC*
*    4. EC outputs are area-averaged per grid cell, and these averages are coupled to the atmospheric component*

*In the following, the EC calculation is described in more detail."*

10. P. 3, Line 27: How is the weight of each elevation class within a grid cell determined?

*The weight of each elevation class within a grid cell is determined by the area of the high-resolution topography dataset that lies within an elevation class. This clarification has been added this clarification to the revised manuscript.*

11. P. 3, Line 28: I believe "average" is referring to the average surface to atmosphere fluxes, and outputs such as SMB and SMB components, but this is not clear.

*This is correct, and text is changed to: "These weights are used to calculate the grid cell average that will be output of CLM4.0 and coupled to CAM4, as well ..."*

12. P. 3, Line 33: Clarify to read "all ECs within a grid cell."

*Changed according to reviewers suggestion.*

13. P. 4, Lines 2-8: It would be helpful if the authors can reiterate here which terms are common to all ECs within a grid cell, and which terms vary by EC as a result of downscaling. In particular it should be mentioned that albedo is calculated interactively within the model for each EC based on snow properties / snow depth over ice.

*This is explained on p3, l31-33. We have added: "Snow albedo is calculated based on snow grain size, depth, density, and other properties."*

14. P. 4, Lines 11-12: Note that these quantities are calculated on the ESM grid. Also, aren't these quantities calculated by the atmospheric model CAM4 and not CLM4?

*These quantities, as mentioned here are calculated by CLM. Take for instance the temperature. It is first calculated in CAM (taken into account e.g. advection), then passed to CLM. CLM simulates exchanges of moisture and heat with the surface, whereafter the temperature is passed on again to CAM.*

15. P. 4, Line 30: Although the details of the setup are described by Vizcaino et al., the authors should mention briefly what forcing is applied (e.g. sea ice/ocean temperatures/ atmospheric nudging).

*Added: "All CESM1.0 model components are allowed to vary freely.".*

*Have added: "*

16. P. 5, Line 15: Make clear why it is necessary to subtract the average CESM grid value from the RACMO2.3 grid cell values. I think this is to only capture gradients within grid cells, and not at the coarser resolution.

*This was to illustrate the scatter as deviations from what is simulated at the grid cell mean, due to different climate realizations in the two models.*

*For clarification, we have added: "We subtract these averages to only capture gradients within each grid cell, and to reduce the effect of internal climate variability."*

17. P. 5, Line 19: "mean elevation" is confusing. Perhaps use "on the CESM grid".

*Changed to: "on the CESM1.0 grid".*

18. P. 5, Line 23: "..comparison of the downscaled SEB components via EC and RCM" is confusing. It is the gradients that are being compared. Revise to something like "…comparison of SEB component gradients for CESM1.0 ECs and the RACMO2.3 RCM."

*Changed accordingly.*

19. P. 6, Line 10: Change "opposite gradient" to "opposite elevation gradient" for clarity.

*Changed accordingly.*

20. P. 6, Lines 16-18: The difference in sign here makes this a bit confusing. Including the longwave components in Fig. 1 or in a supplemental figure would help the reader to easily visualize this.

*We are including this in a supplementary figure.*

21. P. 6, Lines 22-23: Suggest changing "null gradients of incoming radiation in the model and weaker albedo gradients" to "a null gradient of incoming radiation in CESM1.0 and weaker albedo gradients than in RACMO2.3, leading to a smaller gradient in net shortwave radiation." Also, I believe this sentence is only referring to shortwave radiation, but this is not mentioned. Please clarify.

*We do refer to both.*

22. P. 7, Lines 11-12: What is the value of the gradient for CESM1.0?

*The value of the refreezing gradient in CESM1.0 is 62 mm yr$^{-1}$ km$^{-1}$ as mentioned on p7, l3.*

23. P. 7, Lines 14-15: Again, it would be interesting to see the figures for snowfall and refreezing.

*Panel for snowfall is be included in Fig. 2, figure for refreezing is already there (Fig. 2b).*

24. P. 8, Lines 1-3: Not sure what is meant by "non-null variations". It would be clearer to simply note that the albedo gradient increases with increasing lapse rate, as shown in Figure 3.

*It means that more EC points respond to the forcing in form of albedo change.*

25. P. 8, Lines 14-15: Any idea why there is a reversal for the 9.8K/km case?

*It becomes opposite in the 9.8K $km^{-1}$ case as this is stronger than 6 K $km^{-1}$ (whereas 1 K $km^{-1}$ and 4K $km^{-1}$ are weaker) forcing leads to higher amounts of energy being transferred to the lowest areas.*

26. P. 8, Line 19: This is the first time interannual variability is mentioned. Perhaps introduce this with a separate sentence, explaining why interannual variability is interesting in this case and not elsewhere in the study.

*We would argue interannual variability could be interesting in the first part of the study as well. On the other hand, we find the gradients and correlation to be more informative for analyzing the downscaled fluxes, as interannual variability is connected to atmospheric variability which is somewhat taken out when the grid cell mean is subtracted from each EC.*

*See previous answers on internal variability: we have now introduced it in the methods section, where we have also added text to clarify why we do not look at internal variability in the first part.*

27. P. 9, Lines 13-15: This is confusing and should be clarified. I think the authors mean that the mean grid cell elevation is lower than the elevation of the ice sheet, so without ECs, the simulated ice sheet is higher in elevation. This effect was also observed by Alexander et al. (2019).

*Yes. Within each CESM grid cell, there is a range of elevations from the 5 km ice sheet topography. The mean of the elevations from the 5 km topography dataset is higher than the elevation that the CESM atmosphere "sees" both due to smoothing of the atmospheric topography, and that the atmosphere can "see" both ice sheet and vegetated parts of a grid cell. As the lapse rate correction is only applied to elevations corresponding to the ice sheet, this causes a higher areal weight of ice sheet points being forced with a temperature that is lower than the atmosphere simulates. The result of this is cooling of the grid cell.*

28. P. 9, Line 27: Any idea as to why incoming longwave radiation changes?

*Not only does the near-surface heat up, the entire lower part of the atmospheric column warms which is probably leading to the increased incoming longwave radiation.*

29. P. 9, Lines 29-34: It seems ERA-Interim is used for locations outside of the RACMO2.3 domain? Please make this clear. Also specify here which fields are compared.

*ERA-Interim is used for the entire area when calculating the averages for Fig. S1. To clarify, we have added: "Figure S1 compares near-surface temperature, turbulent heat fluxes, net longwave radiation and sea ice extent in EC-1K and EC-6K with ERA-Interim over the entire area …".*

30. P. 9, Line 33: change "as with RACMO2.3" to "as differences with RACMO2.3" for clarity.

*Changed accordingly.*

31. P. 10, Lines 4-5: Other studies (e.g. Vizcaino et al., 2013; Alexander et al., 2019) have evaluated the EC method at the coarse resolution but not at a higher resolution as done here. This should be clarified here.

*Added for clarity.*

32. P. 10, Line 6: "Linear fits" of what? Please clarify.

*Changed to: "These gradients are obtained by linear regression of the components on sub-grid elevations in all GrIS grid cells"*

33. P. 10, Line 22: Change "enables to explore the interaction with" to "enables exploration of the interaction between the high-resolution surface simulation and…"

*Changed accordingly.*

34. P. 10, Line 25: Change "to RCM" to "to the RACMO2.3 RCM".

*Changed accordingly.*

35. P. 11, Line 25: It could be that improving representation of physical processes at the elevation class scale will allow for a better identification of the optimal lapse rate. This could be mentioned here, if the authors agree.

*Yes, that is a good point, and we have added here: "Further improvements of the physical representation of SMB processes at the EC scale might allow for a better identification of an observationally constrained optimal lapse rate."*

36. P. 11, Line 32: Clarify "for radiation", e.g. "for apparent biases in gradients of net radiation"

*Changed to: "compensate for absence of incoming radiation downscaling."*

37. P. 12, Line 11: Clarify "more adequate".

*By adequate we mean to represent snow compaction, firn, refreezing etc. more adequately.*

*Added : "for realistic representation of e.g., snow compaction, firn, and refreezing"*

**Technical corrections**

1. P. 1, Line 2: Remove "the" before "surface mass balance (SMB) modeling"

2. P. 1, Line 4: Change "elevation dependent" to "elevation-dependent"

3. P. 1, Line 20: Change "leading" to "which would lead".

4. P. 1, Line 21: Change "is losing mass" to "has been losing mass".

5. P. 2, Line 9: Perhaps change "seem required" to "are likely required"?

6. P. 2, Line 12: The van Kampenhout paper year can be changed to 2019, with the reference to the final revised paper updated in the reference list

7. P. 2, Line 17: Revise "Statistical downscaling, which uses elevation corrections on…" to "Statistical downscaling uses elevation corrections to…"

8. P. 5, Line 23: Change "r-value" to "r-values".

9. P. 5, Line 25: Change "…solar radiation is not downscaled so that all ECs receive…" to "…solar radiation is not downscaled. As a result, all ECs within a grid cell receive…"

10. P. 6, Line 4: Change "more correlated" to "better correlated".

11. P. 6, Lines 20-21: Change "The net radiation gradients…" to "The net radiation gradient in CESM1.0 is 5.4 W m-2 km—1 and in RACMO2.3 is -22.6 W m-2 km-1 (Table 1)."

12. P. 7, Line 5: Change "low ECs" to "low elevation ECs"

13. P. 7, Line 24: Change "compensates the biases" to "compensates for the biases"

14. P. 7, Line 26: Change "compensates this" to "compensates for this"

15. P. 8, Line 27: Change "therefore not only reflecting" to "therefore reflect more than just"

16. P. 8, Line 30: Change "high ECs" to "high elevation ECs" and "low ECs" to "low elevation ECs"

17. P. 8, Line 34: Change "lower than the magnitude of the respective" to "lower in magnitude than the respective"

18. P. 8, Line 35: Change "gradient is less" to "gradient is also less"

19. P. 9, Line 27: Change "overcompensated by" to "overcompensated for by"

20. P. 10, Lines 11-13: Suggest revising the sentence to read: "However, one of the limitations of comparing with an RCM is that unlike an ESM, the RCM is laterally forced with reanalysis. Also, there are fundamental differences in the physical schemes and simulated climate components between the ESM and RCM compared here."

21. P. 10, Lines 13-14: Change "net longwave" to "net longwave radiation"

22. P. 11, Line 1: Change "although it varies" to "despite the fact that it varies"

23. P. 12, Lines 2-3: Change "efficient to generate" to "efficiently generates"

Thank you, these suggestions are taken into account for the revised manuscript.

**Reviewer #3:**

**General synopsis** This is a useful contribution about the novel application of using Earth System Models (ESM) with downscaling, via sub-gridcell elevation classes, to simulate Greenland Ice Sheet surface mass balance. Although it is fairly model-specific (based on the CESM1.0 ESM), this paper should be of broad interest to the GrIS SMB modelling and Greenland climate communities.

Some previous highly relevant literature is missing or can be better acknowledged (see comments below).

I wonder whether the CESM results can be compared with MAR as well as RACMO, for an independent RCM model check (and since MAR is the main alternative RCM currently used for Greenland)?

**Specific comments**

Page 1, lines 15-16 re. strong Arctic warming: Please add the following reference to those cited: Overland, J.E. and Hanna, Edward and Hanssen-Bauer, I. and Kim, S.-J. and Walsh, J.E. and Wang, M. and Bhatt, U.S. and Thoman, R.L. (2018) Surface air temperature. Arctic Report Card , NOAA.
https://www.arctic.noaa.gov/ReportCard/Report-Card-2018/ArtMID/7878/ArticleID/783/Surface-Air-Temperature

*This reference is added.*

P1, L21 re. "GrIS is losing mass at an accelerated rate": please add the following highly relevant references to those cited: Edward Hanna, Francisco J Navarro, Frank Pattyn, Catia M Domingues, Xavier Fettweis, Erik R Ivins, Robert J Nicholls, Catherine Ritz, Ben Smith, Slawek Tulaczyk, Pippa L Whitehouse, H Jay Zwally (2013) Ice sheet mass balance and climate change. Nature 498, 51-59. Bamber, JL et al. (2018): The land ice contribution to sea level during the satellite era. Environmental Research Letters, 13(6), 063008,

*These references are added.*

P2, L8: should also add there is a significant disparity between different model estimates of GrIS SMB (Fettweis 2018): Fettweis, X. (2018) The SMB Model Intercomparison (SMBMIP) over Greenland: first rlts. AGU Fall Meeting talk archived at:
https://orbi.uliege.be/handle/2268/232923.

Reference for the SMBMIP is added, in addition to: "Also, there is a significant disparity between different model estimates of GrIS SMB even for models with higher resolution"

P2, L19: re. statistical downscaling please add the following highly relevant references: Hanna et al. (2011) AND Wilton et al. (2017) DJ Wilton, A Jowett, E Hanna, GR Bigg, MR Van Den Broeke, X Fettweis, ...(2017) High resolution (1 km) positive degree-day modelling of Greenland ice sheet surface mass balance, 1870–2012 using reanalysis data. Journal of Glaciology 63 (237), 176-193

*This reference is added.*

E Hanna, P Huybrechts, J Cappelen, K Steffen, RC Bales, E Burgess, ...(2011) Greenland Ice Sheet surface mass balance 1870 to 2010 based on Twentieth Century Reanalysis, and links with global climate forcing. Journal of Geophysical Research: Atmospheres 116 (D24)

*This reference is added.*

P2, L32: While the motivation for the study is good as stated, can you make it clear in this sentence/paragraph whether you investigated precipitation downscaling as well as temperature downscaling?

*Yes, we will make it clear by adding "(it must be noted that our model does not downscale precipitation)"*

P3, L26: How was this number of elevation classes chosen? Would having a greater number of classes improve the results?

*The choice was motivated by a compromise between computing time and increased (vertical) resolution. Offline test showed this number is appropriate, and is the default for CESM1.0. We will add this to the revised manuscript.*

P3, L34 "Incoming radiation, precipitation and wind are kept constant across all ECs" - Is this a potential limitation of this study or could improvements be made here?

*This is discussed in detail in section 4 (p.11, l1-11)*

P5, L8 "snow when near-surface temperatures are between -7ˆoC and -1ˆoC" – the latter value (-1C) seems quite a low upper threshold for snow?

*This is the limit of where precipitation falls as snow only, mixed precipitation can occur at higher temperatures. This will be added to the revised manuscript.*

P10, LL4-5 "the first time the EC method for downscaling from a global climate model of ˜100 km to the much higher resolution (5 km) of an ice sheet model" – point out that this kind and magnitude of statistical downscaling has been previously successfully used in downscaling meteorological reanalysis data from ˜100-km resolution to 5-km resolution (Hanna et al. 2005 & 2011, Wilton et al. 2017). E Hanna, P Huybrechts, I Janssens, J Cappelen, K Steffen, A Stephens (2005) Runoff and mass balance of the Greenland ice sheet: 1958–2003. Journal of Geophysical Research: Atmospheres 110 (D13) (Other two references details are above.)

*This reference is added.*

P11, LL9-11: The recommended implementation of a precipitation phase downscaling scheme doesn't really solve the great challenge of overall elevation correction for precipitation. This paragraph therefore sounds a little weak as currently stated – can the authors strengthen their argument here?

*Yes, this wouldn't solve the difficulty of resolving the complex spatial patterns of precipitation over the GrIS. However, we do believe that changing the phase could improve the SMB as it would not allow for e.g., unrealistic rain at higher elevations.*

[revised manuscript text omitted]
) Incoming solar radiation (W m$^{-2}$), b) outgoing solar radiation (W m$^{-2}$), c) incoming longwave radiation (W m$^{-2}$) and d) outgoing longwave radiation (W m$^{-2}$). The lines represent least-squares linear regressions. The annotated $m$ is the least-squares linear regression gradient (mm yr$^{-1}$ km$^{-1}$, $r$ is the correlation coefficient.

[Figure]

**Figure S2.** Annual means of selected climate variables in the simulations EC-1K ("No elevation classes", blue) and EC-6K ("elevation classes", red), and ERA-Interim (only 1979-2005, black) for reference. The data are area-weighted averages (integrated for sea-ice) for the region in Fig. 5.